# Bardsey – an island in a strong tidal stream
## Underestimating coastal tides due to unresolved topography

## J. A. Mattias Green[1,*] and David T. Pugh[2]

[1] School of Ocean Sciences, Bangor University, Menai Bridge, UK
[2] National Oceanography Centre, Joseph Proudman Building, Liverpool, UK
* Corresponding author: Dr Mattias Green, m.green@bangor.ac.uk

_______________________________________

## Abstract

Bardsey Island is located at the western end of the Llŷn Peninsula in north-west Wales Separated from the mainland by a channel some 3 km wide, it is surrounded by reversing tidal streams of up to 4 m s⁻¹ at spring tides. These local hydrodynamic details and their consequences are unresolved by satellite altimetry, nor are they represented in regional tidal models. Here we look at the effects of the island on the strong tidal stream in terms of the budgets for tidal energy dissipation and the formation and shedding of eddies. We show, using local observations and a satellite altimetry constrained product (TPXO9), that the island has a large impact on the tidal stream, and that even in this latest altimetry constrained product the derived tidal stream is under-represented due to the island not being resolved. The effect of the island leads to an underestimate of the current speed in the TPXO9 data in the channel of up to a factor of 2.5, depending on the timing in the spring-neap cycle, and the average tidal energy resource is underestimated by a factor up to 14. The observed tidal amplitudes are higher at the mainland than at the island, and there is a detectable phase lag in the tide across the island – this effect is not seen in the TPXO9 data. The underestimate of the tide in the TPXO9 data has consequences for tidal dissipation and wake effect computation and show that local observations are key to correctly estimate tidal energetics around small-scale coastal topography.

## 1 Introduction

Scientific understanding of global tidal dynamics is well established. Following the advent of satellite observations, up to 15 tidal constituents have been mapped using altimetry constrained numerical models, and the resulting products verified and constrained further using *in situ* tidal data – see Stammer et al. (2014) for details. There is, however, still an issue in terms of spatial resolution of the altimetry constrained products: even the most recent (global) tidal models have only 1/30$^o$ resolution (equivalent to ~3.2 km in longitude at the equator, ~1.9 km in the domain here, and 3.2 km in latitude everywhere). The satellite themselves may have track separation of 100s of km (Egbert and Erofeeva, 2002) and the coastline can introduce biases in the altimetry data which limits the usefulness of it in the assimilation process. Consequently, smaller topographic features and islands are unresolved, and may be "invisible" in altimetry constrained product even if the features may be resolved in the latest bathymetry databases, e.g., the General Bathymetric Chart of the Oceans (GEBCO, https://www.gebco.net/; Jakobsson et al., 2020). This can mean that the energetics in the products, and in other numerical model with insufficient resolution, can be biased because the wakes can act as a large energy sink (McCabe et al., 2006; Stigebrandt, 1980; Warner and MacCready, 2014). Whilst the globally integrated energetics of these models is consistent with astronomical estimates from lunar recession rates (Bills and Ray, 1999; Egbert and Ray, 2001), the local estimates can be wrong. However, new correction algorithms improve the satellite data near coasts (e.g., Piccioni et al., 2018), but this is yet to be included in global tidal products.

Because many of the altimetry constrained tidal databases are models, and not simply altimeter databases, they also provide tidal currents as well as elevations. This is true for TPXO9 (see Egbert and Erofeeva, 2002 and https://www.tpxo.net/ for details), the altimetry constrained product used here. Here, we use a series of tide-gauge measurements from Bardsey Island in the Irish Sea (Figure 1) alongside TPXO9 to evaluate the effect of the island on the tidal dynamics as they track around Bardsey Island. Bardsey Island is a rocky melange of sedimentary and igneous rocks including some granites, located 3.1 km off the Llŷn Peninsula in North Wales, UK (Figure 1a). It is approximately 1 km wide, though only 300 m at the narrowest part, and 1.6 km long. It reaches 167 m at its highest point. Bardsey Sound, between the Llŷn peninsula and the island, experiences strong tidal currents. The relatively small scale of the island and the Sound means that the local detail is not "seen" in the altimetry constrained products. The uncaptured, by the altimetry constrained data, active local tidal dynamics allows us to compare the altimetry constrained tidal characteristics in TPXO9 for the region with accurate local observations and quantify the validity limits of TPXO9 for this type of investigation. We will make a direct comparison of the tidal amplitudes and phases measured by the bottom pressure gauges around the island (see Figure 1b for tide gauge (TG) locations and a summary of the *in situ* tides). We also consider whether, and when, in the tidal cycle, flow separation occurs in the wake of the island.

We will use some basic fluid-flow parameters in our analysis later. Transition to turbulence, and hence flow separation around an object, can be parameterised in terms of a Reynolds number, $Re = UD/v$, where $U$ is a velocity scale, $D$ is the size of the object, and $v{\sim}100$ is a horizontal diffusivity (see, e.g., Wolanski et al., 1984). It indicates when there is a transition to flow separation behind the island: at low Reynolds numbers, $Re<1$, the flow is quite symmetric upstream and downstream, and there is no flow separation at the object. As the Reynolds number is increased to the range $10 < Re <40$, laminar separation happens and results in two steady vortices downstream. As Re increases further, up to $Re<1000$, these steady vortices are replaced by a periodic von Karman vortex street, whereas if $Re>1000$, there is a fully separated turbulent flow (Kundu and Cohen, 2002).

Another useful non-dimensional number for this type of investigation is the Strouhal number, $St = fD/U$. Here, $f$ is the frequency of the shedding of vortices. Fully developed vortices are generated when $T>f$, where $T$ is the frequency of the oscillating flow (Dong et al., 2007; Magaldi et al., 2008). If, on the

other hand, the tidal frequency is larger than $f$ only one wake eddy will be shed on each tidal cycle, if
it has time to form at all.

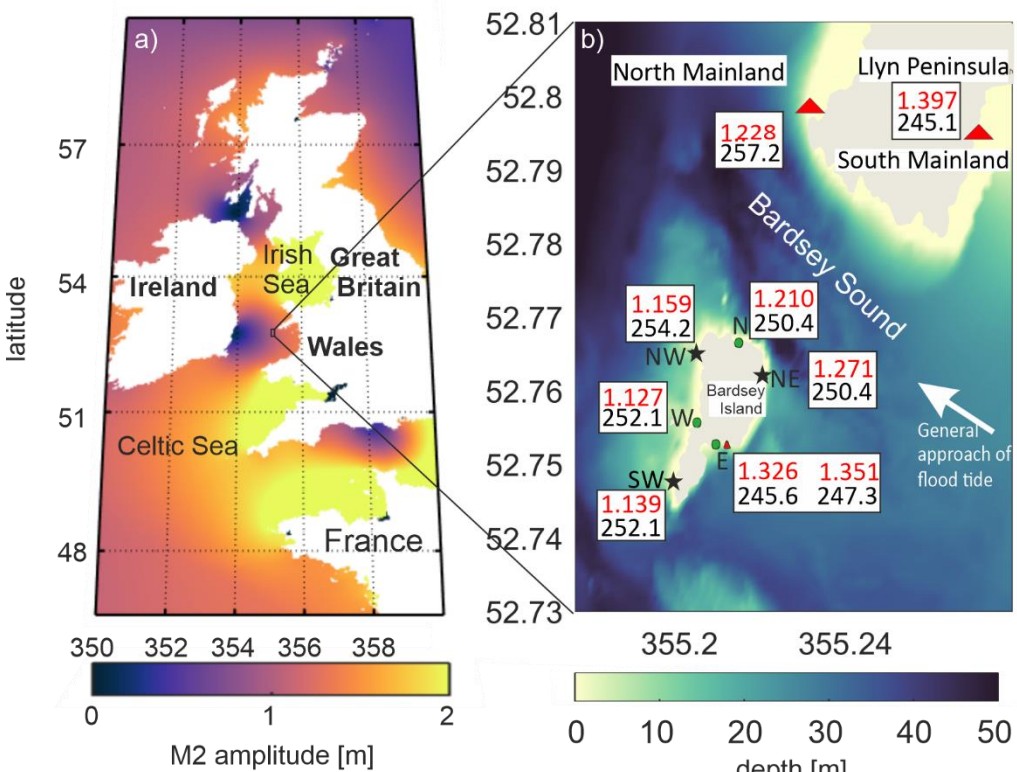

Figure 1: a) Map of the European shelf showing $M_2$ amplitudes in meters, from TPXO9.
b) details of local topography and tidal characteristics in the vicinity of Bardsey Island. The symbols
mark the TG location, with green ellipses denoting Deployment 1, black stars Deployment 2, and red
triangles Deployment 3. Note that East was occupied twice, during Deployments 1 and 3. The red
numbers in the text boxes are the amplitudes (in meters) and the phase lags on Greenwich (in degrees,
one degree is almost two minutes in time) from the harmonic analysis for each tide gauge. The
bathymetry comes from EMODnet (https://www.emodnet-bathymetry.eu/).

## 2 Observations

2.1 *In situ* data collection
The tidal elevations around Bardsey were measured in three Deployments, from summer 2017
through to spring 2018 (Table 1 and Figure 1b). Site East, the main harbour for the island at Y Cafn,
was occupied twice as a control, during Deployment 1 and 3. The other instrument deployments were
bottom mounted a few tens of metres laterally offshore, and all instruments were deployed in depths
between 3.2 m and 16.5 m. The instruments used were RBR pressure recorders with a measurement
resolution better than 0.001 m and they were set to sample every 6 minutes.
The resulting pressure series were analysed to extract tides, using the Tidal Analysis Software Kit of
the National Oceanographic Centre (NOC, 2020). Analyses were made for 26 constituents, including
Mean Sea Level, and eight related constituents, appropriate for a month or more of data (Pugh and
Woodworth, 2014). In Table 2 the three constituents listed are the two biggest, $M_2$ and $S_2$, and (as an
indicator of the presence of shallow water tides) $M_4$, the first harmonic of $M_2$. Shallow water tides  are
enhanced around the island because of the curvature of the flow as it bypasses the island and
headland (see section 6.2.3 of Pugh and Woodworth, 2014). The non-tidal residuals, the final column
in Table 1, compare well with the residuals at Holyhead, the nearest permanent tide gauge station
some 70 km north; for Holyhead these were 0.096 m, 0.172 m, and 0.067 m for the same periods (note
that bottom pressure measurements at Bardsey include a partial natural sea level compensation for
the inverted barometer effect). Deployment 2 residuals at both Bardsey and at Holyhead were
noticeably higher than for the other two Deployments because Deployment 2 included one of the
most severe storms and waves in local memory: hurricane Ophelia, which had maximum local wind
speeds on 16 October 2017. A good indication of the internal quality of the *in situ* observations and
analyses is given by the consistency in the tidal ages and $S_2/M_2$ amplitude ratios. The tidal age is the
time after maximum astronomical tidal forcing and the local maximum spring tides, or approximately
the phase difference between the phases of $S_2$ and $M_2$ in hours, whereas the amplitude ratios are
related to the spring-neap amplitude cycle. These are given in the final columns of Table 2. The effects
of the storm were not noticeable in the tidal signals, as they were at very different natural frequencies.
The subsurface pressure measurements at Bardsey include atmospheric pressure variations, and any
tidal variation therein. However, at these latitudes the atmospheric pressure $S_2$ variations are very
small. At the equator the atmospheric $S_2$ has an amplitude of about 1.25 mb, which decreases away
from the equator as $cos^3$*(latitude)*, so at 53° N the amplitude is reduced to 0.26 mb, a sea level
equivalent of 2.5 mm.
Amplitudes and phases of tidal constituents based on short periods of observations need adjusting to
reflect the long-term values of amplitudes and phases. The values in Table 2 have been adjusted for
both nodal effects and for an observed non-astronomical seasonal modulation of $M_2$. Standard
harmonic analyses include an automatic adjustment to amplitudes and phases of lunar components
to allow for the full 3.7%, 18.6-year modulation due to the regression of lunar nodes. However, the
full 3.7% nodal modulation is generally heavily reduced in shallow water and shelf seas, so local
counter adjustments are needed. The nodal $M_2$ amplitude modulation at Holyhead, the nearest
standard port, is reduced to 1.8% (Woodworth et al., 1991). We have used this value in correcting the
standard 3.7% adjustment. The $M_4$ nodal modulations are twice that for M2. The seasonal $M_2$
modulations are generally observed to have regional coherence, so we have used the seasonal
modulations from 9 years of Newlyn data (in the period 2000-2011). $M_4$ is not seasonally adjusted,
and $S_2$ is not a lunar term, so it is not nodally modulated. These very precise adjustments are possible
and useful, but overall, as stated in the caption to Table 2, for regional comparisons we assume, slightly
conservatively, confidence ranges of 1% for amplitudes and 1.0 degrees for phases.
2.2 TPXO9 data
The altimetry constrained product used in this paper is that of the TPXO9 ATLAS which is derived from
assimilation of both satellite altimeter and tide gauge data (Egbert and Erofeeva, 2002). The resolution
is 1/30° in both latitude and longitude (3.7 km and 2.2 km at Bardsey). We used the elevation and
transport information, and their respective phases, for the $M_2$, $S_2$, and $M_4$ constituents. In the
following calculations, we approximate the largest tidal current speeds or amplitudes as the sum of
the amplitudes of the above three tidal constituents. Of course this is only a crude estimate of the full
Highest and Lowest astronomical tides. Note that we are not allowing for $M_2$ to $M_4$ phase locking, and
the relatively small diurnal tides are ignored. We refer to this as the GA (Greatest Astronomical) in the
following.
2.3 LANDSAT data
Landsat-8 data images were used to identify possible eddies in the currents and further illustrate
unresolved effects due to the island. Note that we are not aiming for a full wake description in this
paper. Data were downloaded from the Earth Explorer website (https://earthexplorer.usgs.gov/).
True colour enhanced RGB images were created with SNAP 7.0 (Sentinel Application Platform;
https://step.esa.int/main/toolboxes/snap/) using the panchromatic band for red (500 - 680nm, 15m
resolution), band 3 for green (530 - 590nm, 30m resolution) and Band 2 for blue (450 - 510 nm, 30m
resolution). The blue and green bands were interpolated using a bicubic projection to the 15m
panchromatic resolution, and brightness was enhanced to allow easier visualization of the wakes. The
images used were taken between 11:00 and 12:00 UTC, when the satellite passed over the area, and
the two images were the only cloud-free ones during the measurement periods that were on different
stages of the tide.
Table 1: Details of the pressure gauge deployments, including non-tidal standard deviations in the sea-
level measurement.

| Station | Latitude North | Longitude East | Time and date Deployed (GMT) | Time and date Recovered (GMT) | Mean Depth (m) | Non-tidal Standard deviation (m) |
|---|---|---|---|---|---|---|
| Deployment 1 | | | | | | |
| North | 52.767 | 355.213 | May 25 2017, 16:05 | July 11 2017, 14:00 | 3.9 | 0.113 |
| East | 52.756 | 355.207 | May 25 2017, 15:57 | July 2017, 13:50 | 7.0 | 0.141 |
| West | 52.753 | 355.202 | May 27 2017, 10:45 | July 5 2017, 11:28 | 5.6 | 0.116 |
| Deployment 2 | | | | | | |
| Northwest | 52.765 | 355.203 | September 1 2017, 00:00 | October 27, 2017, 11:10 | 6.7 | 0.156 |
| Southwest | 52.748 | 355.197 | September 1 2017, 00:00 | October 30, 2017, 11:45 | 7.5 | 0.154 |
| Northeast | 52.762 | 355.220 | September 1 2017, 00:00 | October 30, 2017, 12:40 | 5.5 | 0.150 |
| Deployment 3 | | | | | | |
| East | 52.753 | 355.207 | September 7 2018, 15:12 | October 5, 2018, 09:12 | 3.2 | 0.095 |
| South Mainland | 52.759 | 355.275 | September 7 2018, 13:48 | October 6, 2018, 10:24 | 4.8 | 0.088 |
| North Mainland | 52.781 | 355.236 | September 7 2018, 15:00 | October 7, 2018, 15:12 | 16.5 | 0.083 |


## 3 Results

### 3.1 *In situ* Observations

The results of the tidal harmonic analyses are shown in Table 2. The *in situ* RBR data results are given to
0.001 m and 1.0 degrees. Amplitudes are given to three decimal places as appropriate for the uncertainties
in the RBR data, whereas the timing of constituent phases is probably better than $0.5^{\circ}$ (1 minute in time
for $M_2$). Given the small local tidal differences, it is necessary to consider possible variability among the
RBR tidal constituents across the three deployments, both due to seasonal, and also due to nodal shifts.
Also, there is a statistical uncertainty against background noise, as discussed in Pugh and Woodworth,
(2014), Section 4.6. This statistical uncertainty depends on the estimate of non-tidal noise across the
semidiurnal tidal band, though this can be optimistic as noise may be more sharply focussed at the $M_2$
frequency. In fact, the seasonal uncertainty is most significant here. Based on uncertainties in making the
seasonal and nodal adjustments we conclude that, for regional comparisons we can assume confidence
ranges of 1% for amplitudes and 1.0 degrees for phases. We also note that for station East in 2017,
$M_2+S_2+M_4$ (i.e., our GA) accounts for 93.6% of the tidal variance, with $N_2$, in fourth place, provides
3.7% of the remainder.
A spring-neap cycle of parts of the data from the East and West gauges in Deployment 1 is plotted in
Figure 2 and show a tidal range surpassing 4 m at spring tide. Note that the diurnal constituents are
not discussed further due to their small (<0.1 m) amplitudes. The TG data show $M_2$ amplitudes of 1.210
m (North), 1.347 m (East) and 1.139 m (West, see Table 2). These give pressure gradients around the
island. The East and West sites are separated by 300 m, and the across-island difference in amplitude
give, on spring tides, a level difference of up to 0.5 m. between those two gauges There is also a 6.5°
(13 minutes) phase difference for $M_2$ across the island between East and West, with East leading,
consistent with the tide approaching the island from the south and east and then swinging north and
east around the Llŷn Peninsula headland. Figures 2b-c show the across island level difference plotted
against the measured level at East for two representative days of spring and neap tides, with smaller
differences during neap tides. The plots show that the East levels are some 0.5 metres higher in the
East than on the West, at High Water on spring tides. On neaps the excess is only about 0.3 m. The
differences on the ebb tide are slightly reduced, probably because the direction of flow is partly along
the island, steered by the Llŷn Peninsula.
We do not have access to any current measurements from the region, but the tidal stream is known
to reach up to 4 m s$^{-1}$ in the Sound (Colin Evans, pers. comm., and Admiralty, 2017). There is also a
simple interpretation of the differences in level across the island from East to West, which indirectly
gives approximate values for the wider field of current speeds, which we term, but only in a local
sense, the "far-field" currents. Suppose as an island blocking the tidal stream, and ignoring any side
effects, the pressure head across the island is given solely by the loss of kinetic energy in the flow, by
applying the Bernoulli equation (e.g., Stigebrandt, 1980). The same approach applies for wind forces
on an impermeable fence or wall, and the sea level difference, $\Delta h$, between East and West is then
given as,
$$\Delta h = \frac{v^2}{2g} \hspace{4cm} (1)$$
Here, $v$ is the "far field" tidal current speed and $g$ the gravitational acceleration. Then we may
indirectly compute the "far field" tidal currents from the difference in levels across from East to West
as the tide approaches the island (see Figure 1 for the direction of the oncoming tide). Figure 3 a and
b (red curves) shows the currents so computed, for Day 147 (spring tides) and Day 154 (neap tides),
with the speeds are in metres per second. The black curves are the measured sea levels at East. The
computed "far-field" currents have a maximum over 3 m s$^{-1}$ at springs and around 2 m s$^{-1}$ at neaps,
similar to local estimates (Colin Evans, pers. Comm.). The noise in the level differences, which appears
as noise in the currents (i.e., the red curves), may be an indication of turbulence and eddies discussed
further below.
Along the island the differences between Southwest and North are only a few millimetres for $M_2$,
within the confidence limits on the analyses. This curvature of the streamlines as the flow is squeezed
through Bardsey Sound and swings up around the peninsula, leads to the enhanced generation of non-
linear higher tidal harmonics due to curvature on the reversing tidal stream curves (Pugh and
Woodworth, 2014). This contributes to the large $M_4$ amplitudes around the island and headland (Table
233  2).


Table 2: Results of the tidal (TASK) harmonic analyses. "H" is amplitude (in m) and the phases "G" (degrees relative to Greenwich) are given in italics. The TPXO9 data was interpolated to the TG locations and the resulting data given to 0.01 m. The *in situ* RBR data results are given to 0.001 m and 1.0 degrees. However, for regional comparisons we assume confidence ranges of 1% for amplitudes and 1.0 degrees for phases. RBR constituents are adjusted for nodal and seasonal variations. Amplitudes are given to three decimal places as appropriate for the uncertainties in the RBR data, whereas the timing of constituent phases is probably better than 0.5° (1 minute in time for $M_2$).

| Station | | $M_2$ | | $S_2$ | | $M_4$ | | Tidal Age (hours) | $M_2/S_2$ ratio |
|---|---|---|---|---|---|---|---|---|---|
| | | TG | TPXO | TG | TPXO | TG | TPXO | | |
| DEPLOYMENT 1 | | | | | | | | | |
| North | H | 1.210 | 1.17 | 0.458 | 0.45 | 0.114 | 0.12 | | 0.378 |
| | G | *250.4* | *254.4* | *287.1* | *287.3* | *21.7* | *32.4* | *36.66* | |
| East | H | 1.326 | 1.16 | 0.514 | 0.42 | 0.147 | 0.12 | | 0.387 |
| | G | *245.6* | *253.8* | *283.4* | *286.7* | *49.7* | *34.3* | *37.76* | |
| West | H | 1.139 | 1.15 | 0.434 | 0.42 | 0.138 | 0.12 | | 0.381 |
| | G | *252.1* | *253.7* | *288.4* | *286.6* | *36.1* | *34.8* | *36.26* | |
| DEPLOYMENT 2 | | | | | | | | | |
| NW | H | 1.159 | 1.16 | 0.431 | 0.42 | 0.132 | 0.12 | | 0.372 |
| | G | *254.2* | *254.7* | *287.1* | *287.6* | *36.4* | *33.4* | *32.88* | |
| SW | H | 1.217 | 1.15 | 0.461 | 0.42 | 0.09 | 0.12 | | 0.379 |
| | G | *251.2* | *253.4* | *285.5* | *286.3* | *27.4* | *35.6* | *34.28* | |
| NE | H | 1.271 | 1.15 | 0.482 | 0.43 | 0.096 | 0.12 | | 0.379 |
| | G | *250.4* | *253.8* | *284.0* | *286.7* | *44.0* | *32.8* | *33.58* | |
| DEPLOYMENT 3 | | | | | | | | | |
| East | H | 1.351 | 1.16 | 0.522 | 0.42 | 0.138 | 0.12 | | 0.386 |
| | G | *247.3* | *253.8* | *282.8* | *286.7* | *55.0* | *34.3* | *35.5* | |
| S. Mainland | H | 1.397 | 1.21 | 0.538 | 0.44 | 0.152 | 0.14 | | 0.385 |
| | G | *245.1* | *251.5* | *280.7* | *284.4* | *51.7* | *37.1* | *35.6* | |
| N. Mainland | H | 1.228 | 1.2 | 0.461 | 0.43 | 0.074 | 0.12 | | 0.375 |
| | G | *257.2* | *254.6* | *290.4* | *287.6* | *40.8* | *29.1* | *33.2* | |



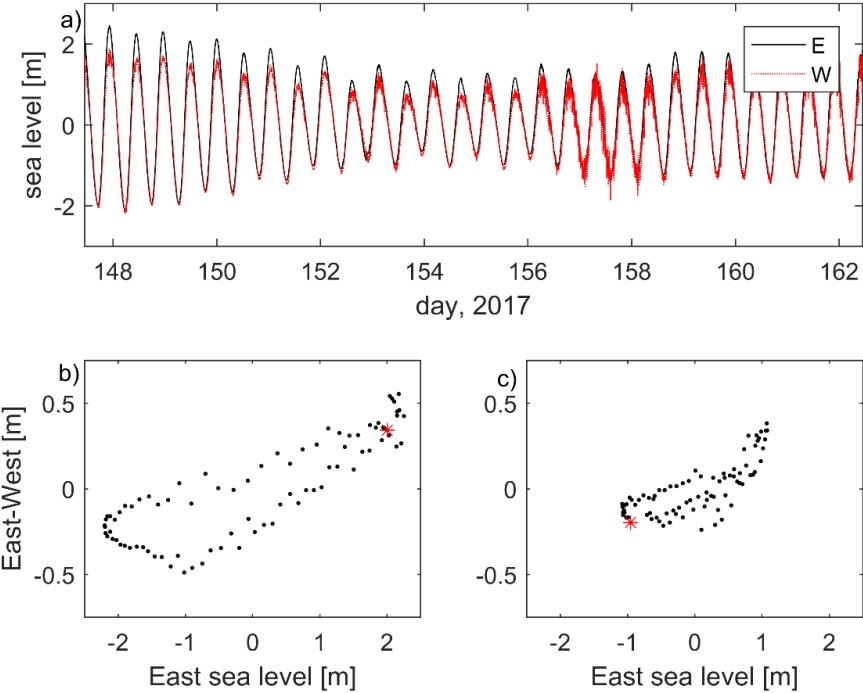

Figure 2: a: Part of the East (black) and West (red) data series, for the in situ data from Deployment 1,
covering one spring-neap cycle (arbitrary datums). b and c: Plots of the East-West elevation difference
vs. the elevation at East for springs (b, day 147) and neaps (c, day 154). The red stars show the data
point for 0000 hours on the day. The progression is clockwise.

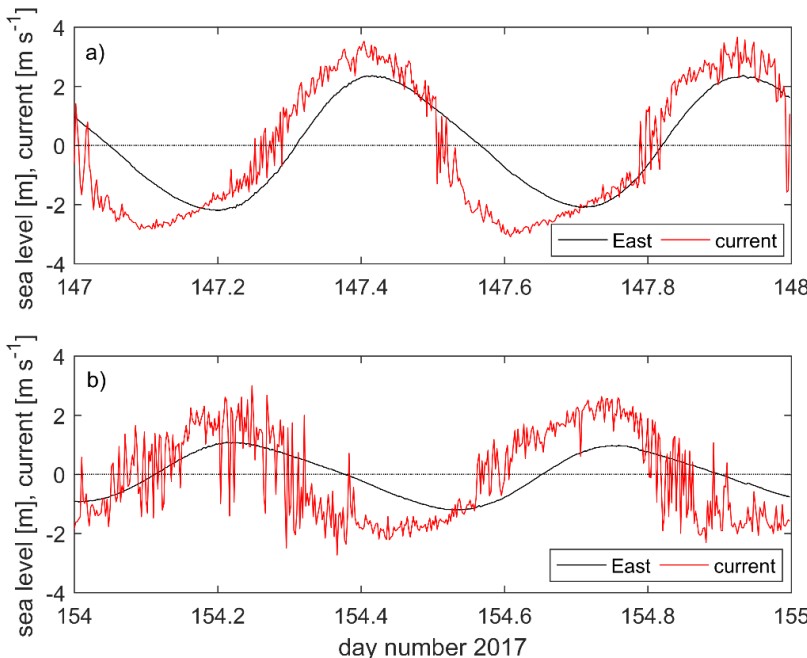

Figure 3: a) Computed current speeds for spring tides, Day 147 (27 May) 2017 in metres per second
(red) compared with the total sea levels at East (in metres, black). The computed currents curve is
noisy as the differences (E-W) are small. The phase relationship between currents is close to a
progressive wave, but with the current maximum to the northwest slightly in advance of the tidal high
water.
b) as in a), but for neap tides on day 154 (4 June) 2017


3.2 Comparison with TPXO9 data
We turn now to a comparison of the tidal analysis data for $M_2$ from the two sources (see Table 2 for
details). When the TPXO9 $M_2$ data, which has no Bardsey Island representation, is interpolated linearly
to the TG positions, the result is only a 0.02 m and 0.7° amplitude and phase difference for the
Deployment 1 locations. Compared to the 0.19 m amplitude difference and 6.5° phase difference in
the TG data, it is clear that there is a substantial deficiency in the TPXO9 model in representing the
role of the island due to its limited resolution. These results are supported by the Deployment 2
measurements (Table 2). Deployment 3 saw an extended and different approach to the data
collection. We revisited East, but also deployed two gauges on the Llŷn peninsula, on the approach to
the island (South Mainland)), and north of it (North Mainland). At South Mainland, TPXO9 is again
underestimating the tidal amplitude by more than 10%. At North Mainland, some 5 km north of
Bardsey, and just north of the Sound, however, the TG and TPXO9 amplitudes are within 1 cm of each
other. This again shows the effect Bardsey and local topography have on the tidal amplitudes in the
region.

As a representation of the shallow-water tidal harmonics, the TPXO9 $M_4$ amplitude agrees well with
the TG data at North (0.12 and 0.11 m, respectively), but overestimates the amplitude at North
Mainland (0.07 m in the TG data and 0.12 m from TPXO; see Table 2). Because higher harmonics are
generated locally by the tidal flow itself, this again shows the effect of the island on the tidal stream;
the M4 amplitude is halved along Bardsey Sound in the TG data, whereas TPXO9 overestimates it and
shows only minor variability. The overestimate in TPXO9 can lead to the tidal energetics being biased
high in the region if they are based on the that data alone.

This is illustrated in the TPXO9 spring and neap flood currents in Figure 4a-b, and the magnitude of
the current in the Sound in Figure 4c. These currents are weaker than the far field estimate using Eq.
(1) above. For spring tides, TPXO9 shows a current of up to 1.5 m s$^{-1}$ in the Sound and 2.5 m s$^{-1}$ in the
far field, whereas the TG data and Eq. (1) comes out at 3.7 m s$^{-1}$ from Eq. (1) for the spring tide far
field (*cf.* Figures 3 and 4). For neaps the corresponding values are 0.6 m s-1 in the Sound and 1.5 m s$^{-}$
$^1$ in the far field from TPXO9, and 3.0 m s$^{-1}$ from the TG data and Eq. (1). The local sea-going experts
(Colin Evans, pers. comm.) and the Admiralty chart for the Sound (Admiralty, 2017) state a current
speed of up 4 m s$^{-1}$, so TPXO9 underestimates the currents in the strait with a factor ~2.5, whereas
the observations, even under the assumptions behind Eq. (1), get within 10%. One can argue that the
sea-level difference along the strait will lead to an acceleration into the strait as well (see e.g.,
Stigebrandt, 1980), that could be added to the far field current. However, frictional effects will come
into play and a large part of the along-strait sea level difference will be needed to overcome friction
and form drag (Stigebrandt, 1980). In fact, of the 0.32 m GA sea-level difference between South and
North Mainland (see Table 1), only 0.006 m is needed to accelerate the spring flow from 3.66 to 4 m
s$^{-1}$ in Eq (1). That means that almost the complete sea-level different along the strait is due to energy
losses.


3.3 Dissipation
The dissipation in a tidal stream can also be computed from $\varepsilon = \rho C_D |u|^3$, where Cd~0.0025 is a drag
coefficient (Taylor, 1920) and $\rho$ = 1020 kg m$^{-3}$ is a reference density. The peak dissipation using the
computed GA current data from Eq. (1) and shown in Figure 3 gives 777 MW for springs and 426 MW
for neaps, assuming the sound is 3.1 km wide and 2.2 km long. This is 0.2-0.4% of the 180 GW of $M_2$
dissipation on the European shelf (see Egbert and Ray, 2000), and is a reasonable estimate for such an
energetic region. Note that this method is independent of the phases between the locations, nor does
it depend on the phases between the amplitudes and currents. If we instead use the  the TPXO9

current speed in the strait, the GA spring dissipation comes out as 53 MW (using $u$ = 1.5 m s$^{-1}$), and the M$_2$ dissipation (using a current speed of 1.2 m s$^{-1}$) as 28 MW. This is an underestimate of a factor 14 for the GA spring tide compared to the computation from the TG data, which again highlights the importance of resolving small-scale topography in local tidal energy estimates, and the use of direct observations in coastal areas to constrain any modelling effort. This dissipation here is only a small fraction of the European Shelf and coastline, but it is a very energetic area. Although the Bardsey tides are unusually energetic, underestimated local coastal energy dissipation may be substantial in the TPXO9 (and similar) data and numerical models.

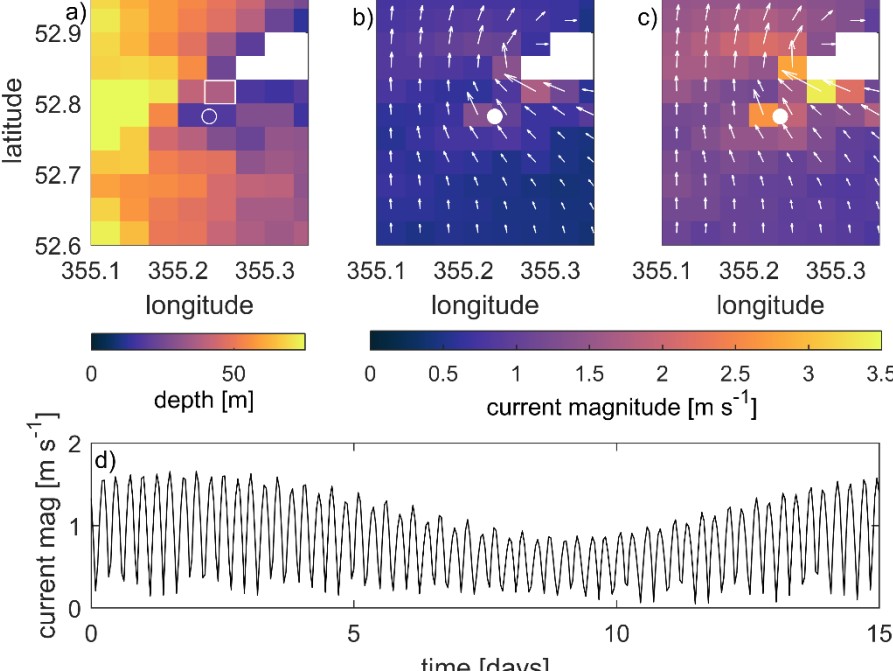

Figure 4: a) The depth from the TPXO9-database covering Bardsey (marked with a white open circle). The rectangle north-west of the island shows the grid cell the data in panel d was extracted from. a)-b) The current magnitude (colour) and vectors at neap (a) and spring (b) flood tides from TPXO9. These are computed from the M$_2$ and S$_2$ constituents only. The white circle shows the location of Bardsey – note that it is not resolved in the TPXO9 data and has been added for visual purposes only. d) The magnitude of the tidal current during a spring-neap cycle in the Sound (i.e., at the cell marked with a rectangle in panel a) using the M$_2$, S$_2$, and M$_4$ constituents in the TPXO9 data. Note that we chose to show data from the centre of the Sound because that is where the computations using Eq. (1) are valid.

### 3.4 Caveat Emptor!

We have shown above that the tidal elevations are underestimated in the TPXO9 data, and that the current magnitude is most likely underestimated as well, so our computations of the energetics and non-dimensional numbers are conservative. The two extremes in tidal current magnitude in Bardsey Sound can be taken to be the neap tide speed from TPXO9 and the GA speed computed using TG data and TPXO9 combined. We thus have 0.9 m s$^{-1}$ (neaps from TPXO9, not discussed above) as the lower range, and 4 m s$^{-1}$ (computed GA) as the upper estimate.

Even using the much-underestimated current speeds from the TPXO-data, the indications are that there would be no stratification locally. The Simpson-Hunter parameter is $\chi = h/u^3 \approx 70$ for Bardsey Sound (Simpson and Hunter, 1974). This means that the area is vertically mixed due to the tides alone. The eddies shed from the island will add more energy to this, further breaking down any potential

stratification from freshwater additions (the Simpson-Hunter parameter is based on heat fluxes only) and act to redistribute sediment. The associated Reynolds number for the Island, $Re = UD/v$, then comes out at approximately 10 for the neap flow, or approximately 40 for the astronomic tidal current (using $D$ = 1000 m as the width and $v$ = 100 m$^2$ s$^{-1}$ as the eddy viscosity). This implies laminar separation into two steady vortices downstream of the Island at peak flows, and the vortices can be expected to appear on both ebb and flood flows (Edwards et al., 2004; Wolanski et al., 1984). There may not be any vortex shedding during neap flows, however, because $Re{\sim}10$.

The Strouhal number $St = fL/U$, is typically about 0.2 for the Re numbers found here (Wolanski et al., 1984), giving $f = St\ U/L = 0.2U/1500 = > 1\text{x}10^{-4} < f < 5\text{x}10^{-4}$ and an associated vortex shedding period of 3-17 hours ($L$ = 1500m is the length of the island). This means that fully developed eddies can be generated at the higher flow rates, because our tidal period (12.4 hours) is longer than the vortex shedding period a few hours). However, at neap flows there is no time to develop a fully separated vortex within the timeframe of a tidal cycle.

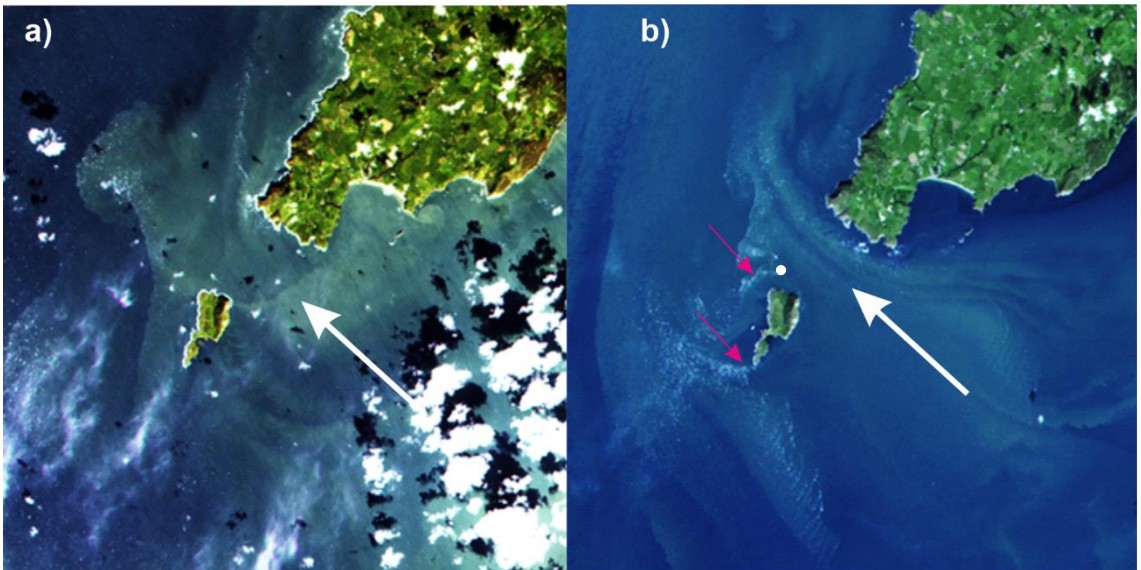

Figure 5: Landsat 8 images from October 5, 2017 (a) and September 13, 2018 (b). The tidal phases are is halfway through the tidal cycle on the neap flood in a) and just after spring high tide in b). The white dot north of the island in panel b is an exposed rock generating a second wake. See https://landsat.gsfc.nasa.gov/data/ for data availability.

This conclusion is supported by satellite images from Landsat 8 (Figure 5), which shows a very different picture between neaps (Figure 5a) and springs (Figure 5b). At spring tides, there are two clear wakes behind the tips of the island (marked with magenta arrows), whereas at neaps (Figure 5a) there is only a more diffuse image in Bardsey Sound, and no signal of a wake behind the south tip of the island.

## 4 Discussion

This brief account was triggered by an interest in detailed mapping of tides in a reversing tidal stream. The results highlight the effect small coastal islands can have on tides in energetic settings, and they highlight the limitations of altimetry-constrained models near coastlines where the bathymetry used in the model is unresolved. Even though TPXO9, which is used here, is constrained by a series of tide gauges in the Irish Sea, including north and south of Bardsey, the island is some 60 km from the nearest long-term tide gauge (in Holyhead, to the north of Bardsey). Consequently, the tidal amplitudes in the database are not representative of the observed amplitudes near the island, and the currents are

underestimated by a factor close to 2.5 for the GA tide. This underestimate also means that wake effects may be underestimated if one relies solely on altimetry constrained models (or coarse resolution numerical models) unable to resolve islands, with consequences for navigation, renewable energy installations, and sediment dynamics.

Future satellite mission may be able to resolve small islands like Bardsey, and improved methods will allow for better detection of the coastlines. In order to obtain tidal currents, however, one still has to assimilate the altimetry data into a numerical model and it will probably be some time before we can simulate global ocean tides at a resolution good enough to resolve an island like Bardsey.

The results do have wider implications for, among others, the renewable industry, because we show that local observations are necessary in regions of complex geometry to ensure the energy resource is determined accurately. Using only TPXO9 data, the dissipation – an indicator of the renewable resource – is underestimating the astronomic potential with a factor up to 14 of the real resource. There is also the possibility that wake effects behind the island would be neglected without proper surveys, leading to an erroneous energy estimate. The results also highlight that concurrent sea level and current measurements are needed to fully explore the dynamics and quantify, e.g., further pressure effects of the island on the tidal stream. Consequently, we argue that in any near-coastal investigation of detailed tidal dynamics, the coastal topography must be explicitly resolved, and any modelling effort should be constrained to fit local observations of the tidal dynamics.

**Acknowledgements:** Instrument deployments and recovery were planned and executed with the assistance of the Bardsey ferry operator, Colin Evans, and by Ernest Evans, the local lobster fisherman and expert on Bardsey tidal conditions. The Deployment 1 obervations were partly funded by the Crown Estate. The Landsat data was processed by Dr Madjid Hadjal and Professor David McKee at University of Strathclyde, and cnstructive comments from Prof. Phil Woodworth and two anonymous reviewers improved the mansucript.

**Code/Data availability:** The data is available from the Open Science Framework (https://osf.io/kvgur/?view_only = ff2d8bd12a61493aa1dfa9011ecdde81)

**Author contributions:** JAMG wrote the manuscript and did the computations. DTP did the measurements, processed the TG data, and assisted with the writing.

**Competing interests:** The authors declare no competing interest

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
