# Peer review of "Bardsey – an island in a strong tidal stream Underestimating coastal tides due to unresolved topography J. A. Mattias Green1,\* and David T. Pugh2 1 School of Ocean Sciences, Bangor University, Menai Bridge, UK 2 National Oceanography Centre, Joseph P"

_Ocean Science, 2020_

## Short Comment (SC1) · 20 Apr 2020

April 2020

Some comments on 'Bardsey – an island in a strong tidal stream Underestimating coastal tides due to unresolved topography' by Green and Pugh

I am not the topical editor or one of the reviewers for this paper, but I gave it a read and have some detailed comments that I hope are useful. I thought it was an interesting paper but the text is not very good and there are many minor problems, especially in the first half. I list these below. I will leave the official reviewers to comment more on

the science.

19, 21, 24, 25 and many other places in the text - there are often mentions of 'altimeter data' or 'altimetry database' but the authors do not use that but instead use the outputs of a hydrodynamic tide model (TPXO9) in which altimeter data (and possibly tide gauge data) have been assimilated. There is a difference between these things and 'altimeter data' is a complete misnomer.

On the other hand, sometimes the language is correct e.g. line 18 'altimetry constrained product'. Fine.

Also everyone knows that altimetry has a coarse spatial (and temporal) sampling and provides elevations and not currents. But on line 14 we read about tidal streams and next line says they will be unresolved by altimetry. Well, yes, of course they will, whatever the spatial resolution.

So I think the text has to be gone through and the misleading language corrected. I suggest that first time you refer to 'altimeter-derived tide model information' (or similar) and thereafter just refer to TPXO9, which is what you mean anyway.

- observations

- ... constituents have been mapped using altimetry

32-34 - it is reasonable (or essential) to say e.g. TPXO here, but pointless to refer to FES and give a web site as you don't use the FES2014 model in the paper and there is no further mention of it below. I suggest that you reword to say e.g. TPXO and several other models and give a reference to Stammer et al. (2014) which the authors will be familiar with.

define TPXO acronym

Also you can add that, because TPXO9 is a model and not a simple altimeter database, it provides tidal currents as well as elevations.

35, 38 - again, there isn't an issue with altimetry products but rather with the models that have used altimetry.

- define GEBCO

50, 51 - ditto the above. I rest my case.

- We will make a ..

- .. for tide gauge (TG) locations ..

- in situ is Latin and has no hyphen, as you use correctly somewhere lower down. You could put it in italics as you do below.

... of the in situ tide gauge measurements).

Figure 1. There are many problems with this:

(i) In (a) can you please change the political Eire and UK to the geographicial Ireland and Great Britain. If you insist on the former then I will insist on you showing the border with Northern Ireland.

(ii) In (b) there is no (b) shown (iii) (b) shows longitudes but not latitudes. Also the caption says 'map data from GE' but there is no bathymetry shown (that would be essential I would have thought, surely you can get bathymetry to 50 metres or so from recent European databases) or land topography so I don't see where GE comes into this.

(iv) in the box for location East, the two sets of amplitudes and phases run into each other with no space.

(v) line 1 of caption meters should be metres as most of the paper has UK English spelling. line 3 - locations. line 3 - drop 'les' line 3 - I can't see any blue crosses. It may be that there are both green and blue dots, I can't tell, but they overlap and you can't see them separately and some people will also have problems telling green from blue.

Also Bardsey Island has no space.

line 4 - Phases should be phases to be consistent with elsewhere. line 5 - amplitudes should be M2 amplitudes, and then it should say 'the black numbers show ...', phases should be Greenwich phase lags and two minutes should be approximately 2 minutes, and 'for each tide gauge' should be 'for each tide gauge record'. Somewhere in the caption one should also refer to Table 1 and the caption should also mention the arrows.

It is important to refer to phase lags instead of phases as (i) they are lags anyway, and (ii) you also use the word phase to refer to a set of measurements.

- I don't think 3.2-16.5 is consistent with 'a few tens' which to me means a much larger number.

- this should read 'using the Tidal Analysis Software Kit of the National Oceanography Centre (NOC, 2020)' and then add NOC (2020) with the web reference to the reference list.

- say when Ophelia was

- in situ

I don't think a reader will automatically understand why the consistency of tidal age (and will he know what that is anyway?) and S2/M2 ratio is important. It could do with some extra words and a reference to Pugh and Woodworth (2014).

Also I felt at this point that there should be a para describing Table 2. The table sort of stands alone and is not really mentioned in the text although there are occasional mentions of it. But a para here would be justified. For example, why did you choose just to show M2, S2 and M4. Then, you are inviting the reader to compare the tide gauge and model values, but S2 is not strictly comparable as the measurements will come from pressure sensors and so include the air tide. You need to mention points like this before the reviewers do.

- 1 minute

Table 1 - column 3 should say East Longitude, 4 should be Time and Date Deployed (hour (GMT), day, month, year), 5 should be Time and Date Recovered (hour (GMT), day, month, year), 6 should be Mean Depth

It is important to spell out the date convention as there is often ambiguity between US and UK conventions.

Phase 2 deployed times have an extra /

- for the reasons explained above I think the title should be Tide Model Information and then the first sentence should read 'The tide model used in this paper is that of the TPXO9 ATLAS which is derived from assimilation of both satellite altimeter and tide gauge data (Egbert and Erofeeva, 2002).'

Actually I was surpised to learn lower down that you say TPXO9 included some tide gauge data as well as altimetry. Well, ok, if that is case the above sentence is needed.

- using the word 'astronomical' in this way is a bit strange. But as you say you are making some kind of analogy with Highest Astronomical Tide. But I wonder if it would better to define some acronym here such as GA to mean 'Greatest Astronomical'.

Also, many times below you refer to astronomic and not astronomical which must be the same thing. Use an acronym instead.

- drop 'we discuss'. reword 'This term is thus analogous to'

- give a reference to SNAP 7.0

- why was this hour chosen and not an hour later for example?

- reword. Summary of findings for M2, S2 and M4 from harmonic tidal analysis of tide gauge and TPXO9 model data. The latter were ... locations given ...

drop 'to ease reading'

Good to have in situ in italics and no hyphen.

Top left of table should be Station

You have TPXO here and in places in the text. It would be best to use TPXO9 throughout.

line 1 of phase 2 has TPXO9 phase to 4 decimal places instead of 2

- phase lags

- .. (west) (Table 2).

I struggled to understand some of the numbers in this para. For one thing why do -14 and -9 have minus signs as you don't specify by difference whether it is east-west or west-east.

Then surely at springs the amplitudes will be larger in the east by 16 cm (8 from M2 and 8 from S2), compared to spring total amplitude of about 1.8 m, which gives 9% to me and not 14.

Then I don't see where the 9% comes from along-island as you don't have a sensor in the south anyway. So please can you spell out things so there is no confusion? Also I don't see where 30 cm comes from - do you mean +/- 16 cm?

150, 151 - phase lags. altimetry data again!

- .. is a substantial model deficiency in representing the role of the island due to its limited resolution, resulting in ..

- drop the comma

I must say I don't find this para very surprising.

- you mean 'As a representation of the shallow-water harmonics, ..'

- altimetry alone. Ditto again.
- you have this the wrong way round. East is on the x-axis so you plotting the difference versus east.

- what does 'the first data point of the day' mean? Do you mean 0 hr on the day.

Figure 3: (i) the colour scale says current amplitude but the caption says current magnitude. I suggest use magnitude for both. Then line 185 says they are spring flood and neap flood but the caption says neap ebb and spring flood. And then because (b) looks to have smaller values anyway I guess that is for neaps? Anyway this is all inconsistent.

- perhaps it would be best to also have the Admiralty chart in the reference list.

- strait. You have called it a sound elsewhere

199, 201 and elsewhere - astronomic - see above

- this is not a suitable heading for a science paper. I suggest you have something like 'Island Tidal Wakes' and by all means express your reservations in the text.

- altimetry data again

- computation of what

- this sentence has no verb

266-269 - this sentence needs rewording. Makes no sense

Figure 4: needs (a) and (b) adding.

- mentions Landsat 8 twice. 273 - is halfway. 274 - 3b should be 4b.

- 3a should be 4a

You say here 4a and 4b are neaps and springs but in the caption says halfway between and after springs.

Also I had to read this twice as from the caption I originally understood that to mean just after a particular spring tide (say an hour after) whereas what I think you mean is after a period of springs (like a day later). Anyway, can you please make this clearer?

- altimetry-constrained models

'where the bathymetry is unresolved' - you mean unresolved in these models. There are in fact decent bathymetry databases available - I suggest you use them for Figure 1(b).

- one is not 'relying solely on altimetry' for the reasons above. You are relying on the models.

- sea level e.g. –> for example reference - please check that you have included them all. Pugh and Woodworth (2014) for example is missing.

---

## Referee Comment (RC1) · Anonymous Referee #1 · 6 May 2020

I think the ms invokes an important issue on the estimation of tidal dissipation using an altimeter-based tidal product, which is widely used worldwide. In my view, the observations were designed with care and conclusions seems to have supported sufficiently by the text. I therefore think the ms would be considered for publication after making technical corrections shown below.

Specific issues: P5L133 Table 2: M2 amp for Stn NE/TPXO (1.5m) is probably 1.15m.

P6L170 Fig.2(a) (this is just a comment and need not to response) In my view, the sea level (especially at site E) seems to show some asymmetric feature, i.e., shorter duration of flood. Is there any possible mechanism leading such a feature?

P7L178 (Fig.3 colour bar legend) I suggest modifying the legend from "current amplitude" to "corrent magnitude", as stated in the figure caption, to avoid a misunderstanding that the property is compiled solely by a single tidal constituent. For the Fig.3 caption, explanation for (a) and (b) is opposite.

P7L199-200 I could not follow how the two figures deltaH=0.07 and Uastro, sm=1.5m/s were deduced (using values in Fig2a?). Please add a brief explanation on this point.

P8L206-207 I suggest removing of a phrase "take the TPXO speed ... as North Mainland, and" to make the context clear. The assumption of using the u_sm was already applied to the discussions developed in the previous paragraph and probably need not to be repeated here.

P8L228 I guess a factor of 0.5 is missing in the definition of the dissipation. In addition, please indicate the actual depth adopted when estimating the dissipation value.

(comment, no need to response) I personally am interested in the impact of the Llyn Peninsula being tilted diagonally (toward NE and SW) against the axis of the Irish Sea and the difference between the main direction of the flood and ebb current around the island indicated, e.g., by Figs.2b and c. This is obviously beyond the range of the current study and looking forward investigating in a near future.

---

## Referee Comment (RC2) · Joanne Williams (Referee) · 26 Jun 2020

Overview: In the manuscript titled "Bardsey – an island in a strong tidal stream", Green and Pugh compare tidal constituents based off of in-situ pressure measurements with constituents derived from a satellite data product. They find that the resolution of the satellite data product is insufficient to accurately describe tidal variations in a small-scale tidal strait. As a result, estimates of tidal dissipation based on the satellite product are biased low. The use of satellite-ocean color measurements to describe the vortex shedding caused by the strong tidal currents in the tidal stream is explored.

Evaluation: Overall, this is an interesting topic and the influence of small scale bathymetry is probably worth bringing to the attention of global modelers and satellite altimetry users. However, the manuscript and the analysis can be improved. For example, the discussion of turbulence and dispersion is vague and can be misinterpreted (see below). The overall framing and importance of the paper can be improved, for example by more specifically discussing why satellite-based or global model based estimates of dissipation matter (see other comments below). Also, a more in depth analysis of the tides is warranted. What are the error statistics on the tidal fit (e.g., RSME) and the uncertainty bounds on the constituents? Did you correct for atmospheric pressure in your in-situ measurements, and does that make a difference (given that a hurricane occurred, maybe it does)? Can tidal statistics derived from 1 month of data be accurately compared to a satellite-based estimate that is obtained from years of sparse data, particularly M4? Perhaps it can, but given the conclusions of the paper this should be explored as an alternate hypothesis for why in-situ and satellite measurements do not agree. Similarly, are there other reasons why a satellite-based estimate may not work well at the coast, beyond resolution? The discussion of tidal velocities—and the comparison to a value in the Admiralty chart—is rather vague. Surely there must be other measurements (e.g., ADCP measurements) or papers, either in reports or the scientific literature? Maybe not, but it is not clear that an exhaustive search has been made to find such values. Similarly, would suggest that authors check that the tidal phase velocity really is sqrt(gh), given their method of estimating dissipation. The use of Landsat is quite qualitative, and could be improved by providing more details and examining many more images (it is not clear whether the figures shown are representative, or just a lucky coincidence). Finally, the manuscript is still a bit 'rough'—in many places, the writing and development of the argument could be made more succinct or focused (see comments below). In addition, the literature reviewed/discussed could/should be expanded (see suggestions below).

**Specific Comments**

Line 14—"some 3 km wide, it is surrounded"—run on sentence. Split into two sentences?

Line 20 "seriously under-represents" is a bit colloquial and vague. Can one be quantitative? "Seriously" is also used later—would suggest rephrasing, here and elsewhere.

Line 23 "at the mainland than at the island"—do you mean near the mainland and near the Island?

Line 31 "several tidal constituents"—How many, and which ones? Would be good to be specific.

Line 34-37—In terms of satellite data analysis, my understanding is that coastal regions have more error. Some of the products out of JPL are specifically tuned to coastal conditions. Perhaps you can comment on some of the near-coast altimetry issues, with references?

General comment: At some point (Introduction? Conclusion?) might be good to mention the new SWOT mission, which has much higher resolution and might make the issues described here obsolete. If so, what lessons might still be used for global tide models (and other global models)? Or, phrased

differently, if global models are not modeling coastal dissipation correctly, how are they (incorrectly) compensating for that in calibration, and what might be the consequences of that?

Line 45—"Rocky mélange"—is this a technical term? Have never seen mélange used outside of novels, but then again I'm not a geologist.

Line 49—Awkward phrasing ("and the separating Sound")

Line 50 "this will lead to effects induced"—what kind of effects? Would be good to be specific

Line 51 would avoid the use of "very". Also, commas would be good here, as in "uncaptured (by TPXO), active, local tidal"

Line 54 "We will do a direct comparison of tidal amplitudes around the island" What kind of comparison? Using what methods? A bit more specificity would be helpful.

Line 60—what are the units on your kinematic viscosity, which equals dynamic viscosity divided by density? Usually this is on the order of magnitude 0.000001 m^2/s, not 100 as mentioned here. Or is "100" a dispersion coefficient? In that case, would seem to be incorrect to call this a kinematic viscosity, in my opinion (even if units are the same). If you are using a diffusion (dispersion) coefficient, which is often based off of a Reynolds number decomposition/gradient diffusion assumption, would also not call this a Reynolds number. Perhaps there is some modifier one can put in front of "Reynolds number", to distinguish it from the usual one. Similarly, wouldn't say this ratio is measuring a transition to turbulence. The flow is turbulent down to a scale of about 1mm (per inertial cascade, to Kolmogorov number). Though I'm not familiar with this "Reynolds number" literature, would assume that this ratio gives some indication of the likelihood of forming large, quasi-2d vortices (what you are calling 'turbulence') vs. having those vortices broken up by dispersive processes (turbulence, shear dispersion, chaotic dispersion….).

Would note that 2D turbulence is much different than 3D turbulence. The implicit assumption you seem to be making is that once the eddies are formed, they are turbulent. Is this strictly speaking correct? The aspect ratio (horizontal to vertical) of these eddies must be very large, where-as in well-developed turbulence energy should be distributed evenly in x,y, and z (not possible due to continuity in a large eddy in a shallow sea). What is the aspect ratio? Might be good to explore and discuss somewhere, and whether it has any implications for the results. How is the evolution of a 2D eddy different from a 3D eddy? How might bottom friction (or sidewall friction) impact the eddy and make it only quasi 2D? In 3D turbulence, there is a cascade of turbulence from large to small scale. In 2D turbulence, that is not the case—energy transfer goes from small to larger scale (e.g., as when small vortices combine to create a larger one). This is not a paper designed to look at such turbulence issues. However, would be good to be more careful in how turbulence is discussed.

Introduction, general: It would be good to briefly review that these small scale 'straits' such as the one being studied are ubiquitous, to frame the larger importance. Angelsey Island in Wales is a nearby example, perhaps. All over the world, there are many Island archipelagos, and some have strong currents such as mentioned here. For example, there is the Greek legend of **Charybdis** , maybe related to currents through the Strait of Messina (Sicily); see https://en.wikipedia.org/wiki/Strait_of_Messina. In Puget Sound, there is Deception Pass (https://en.wikipedia.org/wiki/Deception_Pass ). Within San Francisco Bay, there is Raccoon Strait. Between New York Harbor and Long Island Sound, there is Hells Gate. There are surely many other examples in the world, and some of them may have been studied or at least have references to large currents and whirlpools. Including some information on or review of them may help frame the broader significance of this study.

Introduction, general:  A brief review of diffusion and dispersion might help frame the "viscosity" you use in your "Reynolds number" (assuming my interpretation above is correct).  What is shear dispersion, and is it potentially important here (see for example the book by Fischer et al, from 1979)?  What is chaotic dispersion, and is it important here (see Zimmerman, 1986, and de Swart et al., 1997)?  How can a jet or plume cause horizontal dispersion (e.g., Fong & Stacey, 2003)? What is turbulent diffusion, and is it important here (usually, it's smaller than shear dispersion caused by lateral velocity gradients, but it also depends on the time scale you are considering—shear dispersion becomes effective at larger time scales than turbulent eddy viscosity (and so on).

Introduction, general comment 3:  You could also review the "shallow turbulence" literature, which seems like it might be relevant here.  Uijttewaal & Booij, 2000 and  Uijttewaal & Jirka, 2003 discuss a "shear stability parameter".  Uijttewaal & Booij, 2000 find that eddies produced by lateral shear (*du/dy*) become increasingly suppressed by bottom boundary layer turbulence as depth decreases.  They find that the growth of lateral shear-induced eddies is limited when their shear-stability is greater than approximately 0.1.   Again, it should be noted that 2D turbulence is quite different than 3D turbuluence.  This generally it isn't much considered in shallow coastal waters, or at least I haven't come across it very much.  But maybe there is some more literature since I last thought about it.

Line 89—Did you adjust your pressure measurements for atmospheric pressure variations? If you didn't, would probably be a good idea to do so, just to be complete and make sure that it doesn't significantly alter your analysis.  This is particularly true in your "phase 2" result, in which there was a hurricane.

Line 91 "were subjected to harmonic analysis"—sounds like something unpleasant.  Maybe rephrase, e.g., "were harmonically analyzed"?

Line 96: "residuals have standard deviations appropriate for the region"—this is vague.  Maybe be specific, and compare it to the nearest tide gauge from the same period.

Line 99—" consistency in the tidal ages"  --it might be good to be more specific and define what is meant by 'tidal age', since not all are familiar with this terminology.  Is discussion of tidal age needed?  Some more specificity on what is considered a good fit would help.  Is a good time variation 10 minutes?   1 hour?

Line 109—Does the TPX09 product use the best altimetry product for near coastal areas?  Again, I think JPL has a coastal data product.  Would constituents based off of a coastal data product provide better answers?  One of the main conclusions in the paper is that satellite data have issues in small scale regions.  Is this true of all data products, or just the one used to create the constituent atlas? Another way of putting this—are there other issues, besides resolution, that impact coastal constituents and therefore your comparisons?

General comment:  Would be good to establish somewhere what the typical tidal range in this region is, and that diurnal tidal components are small.  This will help justify the use of only 4 constituents. (Also, is the use of M4 important?  Would be good to establish that quarterdiurnals are important here (or are they)?

Line 114—Would define "Highest and Lowest Astronomical tide" (HAT and LAT), before stating that M2+N2 + S2 +M4 are a limited form.  Also, strictly speaking, M2 and M4 are phase locked, i.e., 2*phaseM2 – phase_M4 = constant (see e.g., Friedrichs & Aubrey 1988).   Unless they have a relative phase of zero, it is incorrect to add their amplitudes together to produce HAT.  Or, rather, one should consider the relative phase when adding.  Is that done here?

Line 117—This is the first mention that I can recall of Landsat.  Why are these images being downloaded?  Leading with a topic sentence that provides some context would be good.

Line 129—The results lead with a table.  I would have expected some text before a table.  Maybe put the table elsewhere?

Amplitudes and phases—Can you think of some way to report confidence intervals or uncertainty, beyond the statement about significant figures?

Line 145—what about frictionally produced overtides?  With a strong current, would seem likely.

Line 148-151—The use of numbers could be reduced and the point made more succinctly, here and elsewhere.  For example, you could say that TPX09 data suggests only a 0.02m and <1 degree difference in M2 in the cross-channel direction, compared to ~0.19m and 6.5 degrees with in-situ data (see Table xxx).  A reader can look at the table for the exact numbers, but doesn't necessarily need to know the exact numbers in the narrative arc (or rather, only needs to know that the TPX gives a much different, and less correct, answer).

Line 162-168:  For someone not familiar with this area, the heavy use of place names is sometimes confusing.

General commnent:  Can one be sure that estimates of M2 and M4 from TPX09 are directly comparable to your one month long measurements, given things like seasonal and interannual variation?  Some discussion and exploration would be good. It seems to me that some review of the TPX analysis would help one frame the results, and help rule out environmentally-based factors as the source of differences in the constitutent analysis.  What is the sampling rate of TPX data, and how long of a data set is needed to obtain good estimates of M2, M4 etc?  Since a long time period is needed, any seasonal variation in tidal constituents are averaged out (see e.g. one of the Mueller papers, or Graewe et al. 2014, or others) . However, the in-situ data would be effected by seasonal effects, and possibly astronomical factors such as the strength of the spring-neap cycle over the measurement month (through frictional interaction). Meteorlogical events like the afforementioned hurricane could also affect M2 and M4, possibly.  One way to look at seasonal cycles would be to evaluate the seasonal cycle in M2 at the nearest long-term tide gauges.  Does such an analysis suggest this a factor in the comparison with TPX?  A seasonal cycle in M2 would produce an M4 variability as well, and therefore any comparison with TPX.  In shallow water, my experience is that M4 can vary a lot from year to year.  TPX constituents are measured over many years, and may therefore "average over" interannual variability.   Other environmental/astronomical variability could also be excluded as a potential factor in your comparison. Does TPX consider the nodal cycle?  Do you adjust for the nodal cycle in in-situ data?

Line 189, Equation 189—What about frictional effects?  Would seem that a fudge factor might be warranted, or perhaps a scaling symbol rather than an equal sign.  In any case, friction is important, and would be good to account for somehow.

Line 191-202—Seems like this paragraph could be reduced in size/explained more succinctly

Line 191-202 – M2 is being used in the scaling equation (Equation 1) and is being compared to a vague maximum velocity of 4m/s.  However, wouldn't the maximum velocity be more likely during a high spring tide, i.e., when the tidal amplitude is caused by M2 +S2 +N2?  Ok, I see this is in the next paragraph.  However, am leaving this comment in, because this paragraph and the next could be presented more succinctly, perhaps together.  Also, would suggest seeing if there are any model or in-situ results in

the peer-reviewed literature than provide estimates of the velocities in this strait, and/or the actual measurements which form the basis for the admiralty charts. The '4 m/s' maximum velocity is quite vague, and the context of this measurement is unknown (was it a wind day? Is it a point measurement, or depth/width averaged? Etc, etc). Therefore, using this value as the gold standard for comparison is a bit iffy.

Line 224—Ok, I see now that friction is being considered. Maybe it would make sense to include all the theory in the Methods section, so that it is more clear that you are considering frictional effects? Note there is no Equation 2 in the manuscript (i.e., Equation 3 is ms-labeled).

Line 226—Did you check that the phase speed really is sqrt(gh)? Since you have the phase progression and know the depth, would be good to check. In shallow water when there is friction and/or convergence, the phase speed can be quite different than sqrt(gh). See e.g., Jay 1991.

Line 226-234—How does this dissipation estimate compare to more local estimates of dissipation, e.g., within the region between England/Wales and Ireland?

Image analysis—how many images were looked at? How representative and statistically significant is the analysis? I would consider looking at more images, to see if the qualitative results are repeatable. For example, you could look at Landsat 7 or Landsat 5 data. You might also consider looking at the ESA Sentinal-3 data as well. It has fantastic resolution and better time resolution than Landsat.

General comment: You might consider looking at Pawlak & MacCready 2001 and Warner & MacCready 2014 for discussion of form drag and eddy formation in the wake of small-scale topography in Puget Sound. Though a stratified region, there might be some useful insights or results in those papers. They also use the Bernoulli Equation, but consider the time-varying potential as well.

General comment: Some more explanation of global models and their resolution is needed. Why is dissipation an important issue? Making this connection will help prove the point that smaller scale resolution can be important.

---

## Editor Comment (EC1) · Joanne Williams (Editor) · 26 Jun 2020

Dear Mattias and David,

I have received a 2nd review for this paper by email, as the reviewer has had some technical difficulties in uploading it. I will upload it as the second review then we can close the discussion. There are quite a few (mostly minor) points to address, and I'm sure you will want to respond to Phil Woodworth's comments as well.

I look forward to reading your response and revision in due course.

Best wishes,

Jo Williams

---

## Author Comment (AC1) · 19 Aug 2020

We are very grateful to the constructive comments from the reviewers and Prof Woodwoorth and we thank them and Dr Williams as editor for their efforts. Our replies to SC1 are included below in bold. We hope they are satisfactory and that the paper is now suitable for publication in OS.

Please also note the supplement to this comment:
https://os.copernicus.org/preprints/os-2020-23/os-2020-23-AC1-supplement.pdf

**Supplement:**

SC1
Some comments on 'Bardsey – an island in a strong tidal stream Underestimating
coastal tides due to unresolved topography' by Green and Pugh

I am not the topical editor or one of the reviewers for this paper, but I gave it a read and have some
detailed comments that I hope are useful. I thought it was an interesting paper but the text is not very
good and there are many minor problems, especially in the first half. I list these below. I will leave the
official reviewers to comment more on the science.

19, 21, 24, 25 and many other places in the text - there are often mentions of 'altimeter data' or
'altimetry database' but the authors do not use that but instead use the outputs of a hydrodynamic
tide model (TPXO9) in which altimeter data (and possibly tide gauge data) have been assimilated.
There is a difference between these things and 'altimeter data' is a complete misnomer. On the other
hand, sometimes the language is correct e.g. line 18 'altimetry constrained product'. Fine.
- **Corrected to "altimetry constrained product" or, more specifically, "TPXO9" throughout.**

Also everyone knows that altimetry has a coarse spatial (and temporal) sampling and provides
elevations and not currents. But on line 14 we read about tidal streams and next line says they will be
unresolved by altimetry. Well, yes, of course they will, whatever the spatial resolution.
- **This sentence (on line 19) has been rewritten: "…**and that even in this latest [TPXO9]
altimetry constrained product the derived tidal stream is seriously under-represented due to
the island not being resolved.**"**

So I think the text has to be gone through and the misleading language corrected. I suggest that first
time you refer to 'altimeter-derived tide model information' (or similar) and thereafter just refer to
TPXO9, which is what you mean anyway.
- **Done when we discuss our results. We have kept "altimetry constrained product" or similar
when discussing the general usability.**

18 – observations
- **Corrected.**

31 - ... constituents have been mapped using altimetry
- **Corrected.**

32-34 - it is reasonable (or essential) to say e.g. TPXO here, but pointless to refer to FES and give a web
site as you don't use the FES2014 model in the paper and there is no further mention of it below. I
suggest that you reword to say e.g. TPXO and several other models and give a reference to Stammer
et al. (2014) which the authors will be familiar with.
- **Rewritten: "**Scientific understanding of global tidal dynamics is well established. Following the
advent of satellite observations, up to 15 tidal constituents have been mapped using altimetry
constrained numerical models, and the resulting products verified and constrained further
using in situ tidal data  – see Stammer et al. (2014) for details.**"**

Define TPXO acronym
- **As far as we are aware TPXO isn't an acronym – it is the name of the database – and there
is not a full name given on the product page.**

Also you can add that, because TPXO9 is a model and not a simple altimeter database, it provides tidal
currents as well as elevations.

- **We have added** "Because many of the altimetry constrained tidal database are models, and not just altimeter databases, they also provide tidal currents as well as elevations. This is true for TPXO9 (see Egbert and Erofeeva, 2002 and https://www.tpxo.net/ for details), the altimetry constrained product used here."

35, 38 - again, there isn't an issue with altimetry products but rather with the models that have used altimetry.
- **This is clarified in the new text:** "…"invisible in altimetry constrained products".

39 - define GEBCO
- **Done**

50, 51 - ditto the above. I rest my case.
- **Corrected in all cases as mentioned above.**

53 - We will make a ..
- **Included**

54 - .. for tide gauge (TG) locations ..
- **Done**

55 - in situ is Latin and has no hyphen, as you use correctly somewhere lower down. You could put it in italics as you do below.
... of the in situ tide gauge measurements).
- **All six instances in the text corrected.**

Figure 1. There are many problems with this:
(i) In (a) can you please change the political Eire and UK to the geographical Ireland and Great Britain. If you insist on the former then I will insist on you showing the border with Northern Ireland.
- **We do not insist – corrected.**
(ii) In (b) there is no (b) shown
- **Added**
(iii) (b) shows longitudes but not latitudes. Also the caption says 'map data from GE' but there is no bathymetry shown (that would be essential I would have thought, surely you can get bathymetry to 50 metres or so from recent European databases) or land topography so I don't see where GE comes into this.
- **Updated with bathymetry.**
(iv) in the box for location East, the two sets of amplitudes and phases run into each other with no space.
- **Fixed.**
(v) line 1 of caption meters should be metres as most of the paper has UK English spelling. line 3 - locations. line 3 - drop 'les' line 3 - I can't see any blue crosses. It may be that there are both green and blue dots, I can't tell, but they overlap and you can't see them separately and some people will also have problems telling green from blue.
- **Rectified, they are now black stars.**
Also Bardsey Island has no space.
- **Fixed**
line 4 - Phases should be phases to be consistent with elsewhere. line 5 – amplitudes should be M2 amplitudes, and then it should say 'the black numbers show ...', phases should be Greenwich phase lags and two minutes should be approximately 2 minutes, and 'for each tide gauge' should be 'for each

tide gauge record'. Somewhere in the caption one should also refer to Table 1 and the caption should also mention the arrows. It is important to refer to phase lags instead of phases as (i) they are lags anyway, and (ii) you also use the word phase to refer to a set of measurements.

- **All suggested corrections done; the caption now reads** "Figure 1: a) Map of the European shelf showing $M_2$ amplitudes in meters, from TPXO9.

b) details of local topography and tidal characteristics in the vicinity of Bardsey Island. The symbols mark the TG location, with green ellipses denoting phase 1, black stars phase 2, and red triangles phase 3. Note that East was occupied twice, during Phases 1 and 3. The red numbers in the text boxes are the amplitudes (in meters) and the phase lags on Greenwich (in degrees, one degree is almost two minutes in time) from the harmonic analysis for each tide gauge. The bathymetry comes from EMODnet (https://www.emodnet-bathymetry.eu/)."

88 - I don't think 3.2-16.5 is consistent with 'a few tens' which to me means a much larger number.

- **This has been clarified – we are referring to the lateral distance as a few tens. It now reads** "The other instrument deployments were bottom mounted a few tens of metres laterally offshore, and in depths between 3.2 m and 16.5 m"

91 - this should read 'using the Tidal Analysis Software Kit of the National Oceanography Centre (NOC, 2020)' and then add NOC (2020) with the web reference to the reference list.

- **Corrected**

98 - say when Ophelia was

- **Added:** "..hurricane Ophelia, which had maximum local wind speeds on 16 October 2017."

99 - in situ

- **Corrected as mentioned above**

I don't think a reader will automatically understand why the consistency of tidal age (and will he know what that is anyway?) and S2/M2 ratio is important. It could do with some extra words and a reference to Pugh and Woodworth (2014). Also I felt at this point that there should be a para describing Table 2. The table sort of stands alone and is not really mentioned in the text although there are occasional mentions of it. But a para here would be justified. For example, why did you choose just to show M2, S2 and M4. Then, you are inviting the reader to compare the tide gauge and model values, but S2 is not strictly comparable as the measurements will come from pressure sensors and so include the air tide. You need to mention points like this before the reviewers do.

- **We have expanded this section to cover all of these points:** "A good indication of the internal quality of the *in situ* observations and analyses is given by the consistency in the tidal ages and $S_2/M_2$ amplitude ratios . The tidal age is the time after maximum astronomical tidal forcing and the local maximum spring tides, or approximately the phase difference between the phases of $S_2$ and $M_2$ in hours, whereas the amplitude ratios are related to the spring-neap amplitude cycle. These are given in the final columns of Table 2. The effects of the storm were not noticeable in the tidal signals, as they were at very different natural frequencies. The subsurface pressure measurements at Bardsey include atmospheric pressure variations, and any tidal variation therein. However, at these latitudes the atmospheric pressure $S_2$ variations are very small. At the equator the atmospheric $S_2$ has an amplitude of about 1.25 mb, which decreases away from the equator as $cos^3(latitude)$ , so at $53^0$ N the amplitude is reduced to 0.26 mb, a sea level equivalent of 2.5 mm. In Table 2 the three constituents listed are the two biggest , $M_2$ and $S_2$ , and as an indicator of the presence of shallow water tides, $M_4$ the first

harmonic of $M_2$ . These shallow water effects are enhanced around the island because of curvature on the directions of current flow.**"**

104 - 1 minute
- **Corrected.**

Table 1 - column 3 should say East Longitude, 4 should be Time and Date Deployed (hour (GMT), day, month, year), 5 should be Time and Date Recovered (hour (GMT), day, month, year), 6 should be Mean Depth It is important to spell out the date convention as there is often ambiguity between US and UK conventions.
Phase 2 deployed times have an extra /
- **All corrections done.**

108 - for the reasons explained above I think the title should be Tide Model Information and then the first sentence should read 'The tide model used in this paper is that of the TPXO9 ATLAS which is derived from assimilation of both satellite altimeter and tide gauge data (Egbert and Erofeeva, 2002).' Actually I was surprised to learn lower down that you say TPXO9 included some tide gauge data as well as altimetry. Well, ok, if that is case the above sentence is needed.
- **This is now more specific than suggested and called "**TPXO9 data**". The regional TPXO Atlas products include some TG data (whereas some is used for validation and error quantification). TPXO9 is a conglomerate including the regional solutions, so the text has been amended to "**The altimetry constrained product  used in this paper is that of the TPXO9 ATLAS which is derived from assimilation of both satellite altimeter and tide gauge data (Egbert and Erofeeva, 2002).**".**

112 - using the word 'astronomical' in this way is a bit strange. But as you say you are making some kind of analogy with Highest Astronomical Tide. But I wonder if it would better to define some acronym here such as GA to mean 'Greatest Astronomical'. Also, many times below you refer to astronomic and not astronomical which must be the same thing. Use an acronym instead.
- **We have opted to rewrite this as "**In the following calculations, we approximate the largest tidal current speeds or amplitudes  as the sum of the amplitudes of the above four tidal constituents. Of course this is only a crude estimate of the full Highest and Lowest astronomical tides. Note that we are not allowing for $M_2$ to $M_4$ phase locking, and the relatively small diurnal tides are ignored. We refer to this as the GA (Greatest Astronomical) in the following.**"**

114 - drop 'we discuss'. reword 'This term is thus analogous to'
- **Done, see above.**

119 - give a reference to SNAP 7.0
- **Done: "**were created with SNAP 7.0 **(**Sentinel Application Platform; https://step.esa.int/main/toolboxes/snap/)**"**

123 - why was this hour chosen and not an hour later for example?
- **Because this was when the satellite passed over the region; Clarified in the text: "**The images used were taken between 11:00 and 12:00 UTC, when the satellite passed over the area.**"**

129 - reword. Summary of findings for M2, S2 and M4 from harmonic tidal analysis of tide gauge and TPXO9 model data. The latter were ... locations given ... drop 'to ease reading'

- **Done and the caption now reads "**Table 1: Results of the tidal (TASK) harmonic analyses. "H" is amplitude (in m) and the phases "G" (degrees relative to Greenwich) are given in italics. The TPXO9 data was interpolated to the TG locations and the resulting data given to 0.01 m. The *in situ* RBR data results are given to 0.001 m and 1.0 degrees. However, for regional comparisons we assume confidence ranges of 1% for amplitudes and 1.0 degrees for phases. RBR constituents are adjusted for nodal and seasonal variations.**"**

Good to have in situ in italics and no hyphen.
Top left of table should be Station
- **added**

You have TPXO here and in places in the text. It would be best to use TPXO9 throughout. line 1 of phase 2 has TPXO9 phase to 4 decimal places instead of 2
- **Corrected to TPXO9 throughout. Decimal places amended.**

135 - phase lags
- **This is now "*In situ* observations"**

137 - .. (west) (Table 2).
- **Included**

I struggled to understand some of the numbers in this para. For one thing why do -14 and -9 have minus signs as you don't specify by difference whether it is east-west or west-east. Then surely at springs the amplitudes will be larger in the east by 16 cm (8 from M2 and 8 from S2), compared to spring total amplitude of about 1.8 m, which gives 9% to me and not 14.
Then I don't see where the 9% comes from along-island as you don't have a sensor in the south anyway. So please can you spell out things so there is no confusion? Also I don't see where 30 cm comes from - do you mean +/- 16 cm?

- **We have corrected and clarified the paragraph: "**The results of the tidal harmonic analyses are shown in Table 2. A spring-neap cycle of parts of the data from the East and West gauges in Phase 1 is plotted in Figure 2a. The TG data show amplitudes of 1.210 m (North), 1.347 m (East) and 1.139 m (West, see Table 2). These give pressure gradients around the island. The narrowest part of the island, some 300 m separates the East and West sites. Here, across-island difference in amplitude give, on spring tides a level difference across 300 m of up to 0.5 m. There is also 6.5° (13 minutes) phase difference for $M_2$ across the island between the east and the west, with the east leading, consistent with the tide approaching the island from the south and east and then swinging north and east around the Llŷn Peninsula headland. Figures 2 b, c show the across island level difference plotted against the measured level at East for two representative days of spring and neap tides. Obviously, the differences are smaller for neap tides. The plots show that the East levels are some 0.5 metres higher in the East than on the West, at High Water on spring tides. On neaps the excess is only about 0.3 m. The differences on the ebb tide are slightly reduced, probably because the direction of flow is partly along the island, steered by the Llŷn Peninsula.**"**

150, 151 - phase lags. altimetry data again!
- **Corrected.**

152 - .. is a substantial model deficiency in representing the role of the island due toits limited resolution, resulting in ..

- **Corrected: "**is a substantial deficiency in the TPXO9 model in representing the role of the island due to its limited resolution, resulting in  a 13% difference in amplitude between the TPXO and TG data at station East. "

159 - drop the comma
I must say I don't find this para very surprising.
- **Comma corrected and the lack of surprise noted.**

162 - you mean 'As a representation of the shallow-water harmonics, ..'
- **Indeed, rewritten as suggested.**

168 - altimetry alone. Ditto again.
- **Corrected.**

173 - you have this the wrong way round. East is on the x-axis so you plotting the difference versus east.
174 - what does 'the first data point of the day' mean? Do you mean 0 hr on the day.
**The caption is rewritten; "**Plots of the East-West elevation difference *vs. the* elevation at East for springs (b, day 147) and neaps (c, day 154). The red stars show the data point for 0000 hours on the day. The progression is clockwise.**"**

Figure 3: (i) the colour scale says current amplitude but the caption says current magnitude. I suggest use magnitude for both. Then line 185 says they are spring flood and neap flood but the caption says neap ebb and spring flood. And then because (b) looks to have smaller values anyway I guess that is for neaps? Anyway this is all inconsistent.
- **We opted to change the caption to** "amplitude". **The figures show spring flood and neap ebb and the text has been corrected. The full caption is now** "Figure 2: The current amplitude (colour) and vectors at spring flood (a) and neap ebb(b) from TPXO9. The white circle shows the location of Bardsey – note that it is not resolved in the TPXO9 data and has been added for visual purposes only. "

184 - perhaps it would be best to also have the Admiralty chart in the reference list.
- **Added.**

194 - strait. You have called it a sound elsewhere
- **Corrected.**

199, 201 and elsewhere - astronomic - see above
- **Corrected throughout to "**GA tide**" as suggested above.**

244 - this is not a suitable heading for a science paper. I suggest you have something like 'Island Tidal Wakes' and by all means express your reservations in the text.
- **We disagree with this, and the formal reviewers have not raised concerns, so we will keep it as it is.**

245 - altimetry data again
- **Corrected.**

246 - computation of what

- **Changed to "…**so our computations of the energetics and non-dimensional numbers are conservative.**"**.

253 - this sentence has no verb
- **Now it does – rewritten as "**The Simpson-Hunter parameter is $X = h/u^3 \approx 70$ for Bardsey Sound (Simpson and Hunter, 1974).**"**.

266-269 - this sentence needs rewording. Makes no sense
- **Rewritten as "**This means that fully developed eddies can be generated at the higher flow rates, because our tidal period (12.4 hours) is longer than the vortex shedding period a few hours). However, at neap flows there is no time to develop a fully separated vortex within the timeframe of a tidal cycle.**"**

Figure 4: needs (a) and (b) adding.
- **Added**

272 - mentions Landsat 8 twice. 273 - is halfway. 274 - 3b should be 4b.
- **Corrected: "**Figure 4: Landsat 8 images from October 5, 2017 (a) and September 13, 2018 (b) from Landsat 8. The tidal phases are halfway through the tidal cycle on the neap flood in a) and just after spring high tide in b)….**"**

278 - 3a should be 4a
- **Corrected**

You say here 4a and 4b are neaps and springs but in the caption says halfway between and after springs. Also I had to read this twice as from the caption I originally understood that to mean just after a particular spring tide (say an hour after) whereas what I think you mean is after a period of springs (like a day later). Anyway, can you please make this clearer?
- **Clarified, see above.**

286 - altimetry-constrained models
'where the bathymetry is unresolved' - you mean unresolved in these models. There are in fact decent bathymetry databases available - I suggest you use them for Figure 1(b).
- **Rewrtitten: "…**highlight the limitations of altimetry-constrained models near coastlines where the bathymetry used in the model is unresolved…**".**

292 - one is not 'relying solely on altimetry' for the reasons above. You are relying on the models.
- **Corrected to "…**altimetry constrained models…**"**

301 - sea level
- **Corrected**

302 e.g. –> for example
- **Our preferred use is e.g., and we have kept that.**

reference - please check that you have included them all. Pugh and Woodworth (2014) for example is missing.
- **Corrected**

---

## Author Comment (AC2) · 19 Aug 2020

We are very grateful to the constructive comments from the reviewers and Prof Woodwoorth and we thank them and Dr Williams as editor for their efforts. Our replies to RC1 are included below in bold. We hope they are satisfactory and that the paper is now suitable for publication in OS.

Please also note the supplement to this comment:
https://os.copernicus.org/preprints/os-2020-23/os-2020-23-AC2-supplement.pdf

[Figure]

**Supplement:**

General reply

We are very grateful to the constructive comments from the reviewers and Prof Woodwoorth and we thank them and Dr Williams as editor for their efforts. Our replies are included below in bold. We hope they are satisfactory and that the paper is now suitable for publication in OS.

RC1

I think the ms invokes an important issue on the estimation of tidal dissipation using an altimeter-based tidal product, which is widely used worldwide. In my view, the observations were designed with care and conclusions seems to have supported sufficiently by the text. I therefore think the ms would be considered for publication after making technical corrections shown below.

- **We thank the reviewer for their constructive comments and hope that they find the revised version of the manuscript suitable for publication.**

Specific issues:

P5L133 Table 2: M2 amp for Stn NE/TPXO (1.5m) is probably 1.15m.

- **Indeed, thank you for spotting. Corrected.**

P6L170 Fig.2(a) (this is just a comment and need not to response) In my view, the sea level (especially at site E) seems to show some asymmetric feature, i.e., shorter duration of flood. Is there any possible mechanism leading such a feature?

- **Yes, it is a tidal asymmetry due to frictional effects and represented by higher harmonics in the harmonic analysis.**

P7L178 (Fig.3 colour bar legend) I suggest modifying the legend from "current amplitude" to "corrent magnitude", as stated in the figure caption, to avoid a misunderstanding that the property is compiled solely by a single tidal constituent. For the Fig.3 caption, explanation for (a) and (b) is opposite.

- **Both issues have been corrected (see also the reply to Prof. Woodworth's short comment): "**The current magnitude (colour) and vectors at spring (a) and neap (b) flood tides from TPXO9. These are computed from the M2 and S2 constituents only. The white circle shows the location of Bardsey – note that it is not resolved in the TPXO9 data and has been added for visual purposes only.
- c) The magnitude of the tidal current during a spring-neap cycle in the Sound using the M2, S2, and M4 constituents in the TPXO9 data."**.**

P7L199-200 I could not follow how the two figures deltaH=0.07 and Uastro, sm=1.5m/s were deduced (using values in Fig2a?). Please add a brief explanation on this point.

- **This is indeed confusing and this paragraph has been rewritten in the light of the new far-field current calculations: "** This is illustrated in the TPXO9 spring and neap flood currents in Figure 4a-b, and the magnitude of the current in the Sound in Figure 4c. These currents are weaker than the far field estimate using Eq. (1) above. For spring tides, TPXO9 shows a current of up to 1.5 m s-1 in the Sound and 2.5 m s-1 in the far field, whereas the TG data and Eq. (1) comes out at 3.7 m s-1 from Eq. (1) for the spring tide far field (cf. Figures 3 and 4). For neaps the corresponding values are 0.6 m s-1 in the Sound and 1.5 m s-1 in the far field from TPXO9, and 3.0 m s-1 from the TG data and Eq. (1). The local sea-going experts (Colin Evans, pers. comm.) and the Admiralty chart for the Sound (Admiralty, 2017) state a current speed of up to 8 knots, or 4 m s-1, so TPXO9 underestimates the currents in the strait with a factor ~2.5, whereas the observations, even under the assumptions behind Eq. (1), get within 10%. One can argue that the sea-level difference along the strait will lead to an acceleration into the strait as well (see e.g., Stigebrandt, 1980), that could be added to the far field current. However, frictional effects will come into play and a large part of the along-strait sea level difference will be needed to overcome friction and form drag (Stigebrandt, 1980). In fact, of

the 0.32 m GA sea-level difference between South and North Mainland (see Table 1), only 0.006 m is needed to accelerate the spring flow from 3.66 to 4 m s-1 in Eq (1). That means that almost the complete sea-level different along the strait is due to energy losses.**"**

P8L206-207 I suggest removing of a phrase "take the TPXO speed … as North Mainland, and" to make the context clear. The assumption of using the u_sm was already applied to the discussions developed in the previous paragraph and probably need not to be repeated here.

**This has also been rewritten using the new current estimates and the TPXO currents instead. The paregarph now read "**To first order, dissipation can be computed from the TPXO9 speed and from the observed amplitude drop along the Sound by comparing the tidal energy flux, $E_f$, between the two locations. A decrease in the energy flux between two locations can be associated with local dissipation of tidal energy as the wave propagates them (see e.g., Green et al., 2008). The flux of tidal energy is given by (e.g., Phillips, 1977)

$$E_f = 0.5 c_g \rho g H^2 \tag{3},$$

where H is again the tidal amplitude and $c_g = \sqrt{gh}$ is the speed of the tidal wave (*h* is the water depth in the Sound, taken to be 37 m), and ρ=1020 kg m-3 is a reference density. The dissipation, $\varepsilon$, is then the difference in energy flux between the two mainland TG locations, or $\varepsilon = 0.5 c_g \rho g (H_{SM}^2 - H_{NM}^2)$, taking $c_g$ constant because h changes little between the TG locations. Using the TG amplitudes, the GA tide would then dissipate 119 kW m$^{-1}$. Over the 3.1 km width of the Sound, this integrates to 368 MW. The M2 tide contributes 31% of this, or 131 MW. This is approximately 0.06% of the total M$_2$ dissipation on the European shelf estimated from TPXO9 (see also Egbert and Ray, 2000), and is a reasonable estimate for such an energetic region. Note that this method is independent of the phases between the locations, nor does it depend on the phases between the amplitudes and currents.

The dissipation in a tidal stream can also be computed from $\varepsilon = \rho C_D |u|^3$, where Cd~0.0025 is a drag coefficient (Taylor, 1920). Using the TPXO9 current speed in the strait, assuming the Sound to be 3.1 km wide and 2 km long, the GA spring dissipation comes out as 35 MW (u-1.5 m s-1), and the M$_2$ dissipation (using a current speed of 1.2 m s-1) as 23 MW. This is a substantial underestimate (factors of 10 and more than 6, for the GA and M$_2$ tides, respectively), which again highlights the importance of resolving small-scale topography in local tidal energy estimates, and the use of direct observations in coastal areas to constrain any modelling effort. This dissipation here is only a small fraction of the European Shelf and coastline, and although the Bardsey tides are unusually energetic, underestimated local coastal energy dissipation may be substantial in the TPXO9 (and similar) data and numerical models.**"**

P8L228 I guess a factor of 0.5 is missing in the definition of the dissipation. In addition, please indicate the actual depth adopted when estimating the dissipation value.
   - **See reply above – corrected.**

(comment, no need to response) I personally am interested in the impact of the Llyn Peninsula being tilted diagonally (toward NE and SW) against the axis of the Irish Sea and the difference between the main direction of the flood and ebb current around the island indicated, e.g., by Figs.2b and c. This is obviously beyond the range of the current study and looking forward investigating in a near future.
   - **We agree, there's a lot more to be done on this topic. Note, however that the tidal stream hits the island's broadside (see Figure 1).**

---

## Author Comment (AC3) · 19 Aug 2020

**General reply**

We are very grateful to the constructive comments from the reviewers and Prof Woodwoorth and we thank them and Dr Williams as editor for their efforts. Our replies are included below in bold. We hope they are satisfactory and that the paper is now suitable for publication in OS.

**RC2**

Overview: In the manuscript titled "Bardsey – an island in a strong tidal stream", Green and Pugh compare tidal constituents based off of in-situ pressure measurements with constituents derived from a satellite data product. They find that the resolution of the satellite data product is insufficient to accurately describe tidal variations in a small-scale tidal strait. As a result, estimates of tidal dissipation based on the satellite product are biased low. The use of satellite-ocean color measurements to describe the vortex shedding caused by the strong tidal currents in the tidal stream is explored.

**Evaluation:**

Overall, this is an interesting topic and the influence of small scale bathymetry is probably worth bringing to the attention of global modelers and satellite altimetry users. However, the manuscript and the analysis can be improved. For example, the discussion of turbulence and dispersion is vague and can be misinterpreted (see below). The overall framing and importance of the paper can be improved, for example by more specifically discussing why satellite-based or global model based estimates of dissipation matter (see other comments below). Also, a more in depth analysis of the tides is warranted.

- We thank the reviewer for these suggestions, and we have endeavoured to better motivate the study and explain why dissipation matters in the revised manuscript. The tidal analysis is now extended as well.

What are the error statistics on the tidal fit (e.g., RSME) and the uncertainty bounds on the constituents?

- These have been added to the paper. The measurements are accurate to fractions of a cm, and we have added this to the text.

Did you correct for atmospheric pressure in your in-situ measurements, and does that make a difference (given that a hurricane occurred, maybe it does)?

No, because it doesn't matter for tides; the harmonic analysis is frequency specific and will ignore the storm effects; we have added "The effects of the storm were not noticeable in the tidal signals, as they were at very different natural frequencies. The subsurface pressure measurements at Bardsey include atmospheric pressure variations, and any tidal variation therein. However, at these latitudes the atmospheric pressure S2 variations are very small. At the equator the atmospheric S2 has an amplitude of about 1.25 mb, which decreases away from the equator as [cos] ^3 (latitude), so at 530 N the amplitude is reduced to 0.26 mb, a sea level equivalent of 2.5 mm. In Table 2 the three constituents listed are the two biggest, M2 and S2, and (as an indicator of the presence of shallow water tides) M4, the first harmonic of M2. These shallow water effects are enhanced around the island because of curvature on the directions of current flow."

Can tidal statistics derived from 1 month of data be accurately compared to a satellite-based estimate that is obtained from years of sparse data, particularly M4? Perhaps it can, but given the conclusions of the paper this should be explored as an alternate hypothesis for why in-situ and satellite measurements do not agree. Similarly, are there other reasons why a satellite-based estimate may not work well at the coast, beyond resolution?

- We argue that it can. After all, the observations really show what the tide was doing, to a very high accuracy. They don't agree, so if you had used TPXO9 to estimate the tides for the

area you would miss out on about NN% of the signal. Land interferes with the actual data return from the satellite because their footprint is quite large. But the altimetry data is assimilated into a numerical model solution, so we feel that the resolution of that model nautical miles at best \_ is certainly large source 2 а of error. The following paragraph is now in the text to justify this: "Amplitudes and phases of tidal constituents based on short periods of observations need adjusting to reflect the long term values of amplitudes and phases. The values in Table 2 have been adjusted for both nodal effects and for an observed non-astronomical seasonal modulation of M2. Standard harmonic analyses include an automatic adjustment to amplitudes and phases of lunar components to allow for the full 3.7%, 18.6 year modulation due to the regression of lunar nodes. However, the full 3.7% nodal modulation is generally significantly reduced in shallow water and shelf seas, so local counter adjustments are needed. The nodal M2 amplitude modulation at Holyhead, the nearest standard port, is reduced to 1.8% (Woodworth et al., 1991). We have used this value in correcting the standard 3.7% adjustment. The M4 nodal modulations are twice that for M2. The seasonal M2 modulations are generally observed to have regional coherence, so we have used the seasonal modulations from 9 years of Newlyn data (in the period 2000-2011). M4 is not seasonally adjusted, and S2 is not a lunar term, so is not modulated nodally. These very precise adjustments are possible and useful, but overall as stated in the caption to Table 2, for regional comparisons we assume, slightly conservatively, confidence ranges of 1% for amplitudes and 1.0 degrees for phases."

The discussion of tidal velocities—and the comparison to a value in the Admiralty chart—is rather vague.

**- This has been rewritten – see reply to RC 1.**

Surely there must be other measurements (e.g., ADCP measurements) or papers, either in reports or the scientific literature? Maybe not, but it is not clear that an exhaustive search has been made to find such values. Similarly, would suggest that authors check that the tidal phase velocity really is sqrt(gh), given their method of estimating dissipation.

- Not that we are aware of, and we did shop around. A comment has been added about the phase relationship.

The use of Landsat is quite qualitative, and could be improved by providing more details and examining many more images (it is not clear whether the figures shown are representative, or just a lucky coincidence).

Yes, it is because we brought these into illustrate to wider implications, not to make a full analysis of the wakes. We have clarified in the text that these are the only images during the measurement periods where Bardsey is visible. This is the reason for why the discussion is brief. Also, focus of the paper is really the effect on the tidal stream itself, rather than downstream effects. This has been clarified: "Landsat-8 data images were used to identify possible eddies in the currents and further illustrate unresolved effects due to the island. Note that we are not aiming for a full wake description in this paper. ... and the two images were the only cloud-free ones during the measurement periods that were on different stages of the tide."

Finally, the manuscript is still a bit 'rough'—in many places, the writing and development of the argument could be made more succinct or focused (see comments below). In addition, the literature reviewed/discussed could/should be expanded (see suggestions below).

- The text has been reworked based on the comments from all reviewers.

Specific Comments

Line 14—"some 3 km wide, it is surrounded"—run on sentence. Split into two sentences?

- Done

Line 20 "seriously under-represents" is a bit colloquial and vague. Can one be quantitative? "Seriously" is also used later—would suggest rephrasing, here and elsewhere.

- Done throughout.

Line 23 "at the mainland than at the island" - do you mean near the mainland and near the Island?

- No, we mean the observations, taken very close the coast, and don't think this warrants rewriting.

Line 31 "several tidal constituents"—How many, and which ones? Would be good to be specific.

- Good point, this now reads . Following the advent of satellite observations, up to 15 tidal constituents have been mapped using altimetry constrained databases..." We see little point in listing the 15 constituents – they are provided on the TPXO9 webpage.

Line 34-37—In terms of satellite data analysis, my understanding is that coastal regions have more error. Some of the products out of JPL are specifically tuned to coastal conditions. Perhaps you can comment on some of the near-coast altimetry issues, with references?

- **Comment added (see above as well): "**However, new correction algorithms improve the satellite data near coasts (e.g., Piccioni et al., 2018), but this is yet to be included in global tidal products."

General comment: At some point (Introduction? Conclusion?) might be good to mention the new SWOT mission, which has much higher resolution and might make the issues described here obsolete. If so, what lessons might still be used for global tide models (and other global models)? Or, phrased differently, if global models are not modeling coastal dissipation correctly, how are they (incorrectly) compensating for that in calibration, and what might be the consequences of that?

Fair comment, and this has been added to the discussion: "Future satellite mission may be able to resolve small islands like Bardsey, and improved methods will allow for better detection of the coastlines. In order to obtain tidal currents, however, one still has to assimilate the altimetry data into a numerical model and it will probably be some time before we can simulate global ocean tides at a resolution good enough to resolve an island like Bardsey. ". As for the dissipation: it will probably be slightly overestimated in the offshore cells and underestimated closer to the coast, so the regional total is most likely accurate. This has not been added to the paper, however.

Line 45—"Rocky mélange"—is this a technical term? Have never seen mélange used outside of novels, but then again l'm not a geologist.

It is a correct geological term describing a large-scale breccia. No changes have been made.

Line 49—Awkward phrasing ("and the separating Sound")

- Rewritten: "...island and the Sound means..."

Line 50 "this will lead to effects induced" — what kind of effects? Would be good to be specific

- **Clarified:** "This will lead to the effects of the island on the tidal stream, e.g., flow acceleration/blocking and wake effects, being missed in those products."

Line 51 would avoid the use of "very". Also, commas would be good here, as in "uncaptured (by TPXO), active, local tidal"

- Amended: "The uncaptured, by the altimetry constrained data, active local tidal dynamics"

Line 54 "We will do a direct comparison of tidal amplitudes around the island" What kind of comparison? Using what methods? A bit more specificity would be helpful.

- Yes, this is vague and now reads "We will make a direct comparison of the tidal amplitudes and phases measured by the bottom pressure gauges around the island (see Figure 1b for tide gauge (TG) locations and a summary of the *in situ* tides). We also consider whether, and when, in the tidal cycle, flow separation occurs in the wake of the island."

Line 60—what are the units on your kinematic viscosity, which equals dynamic viscosity divided by density? Usually this is on the order of magnitude 0.000001 m^2/s, not 100 as mentioned here. Or is "100" a dispersion coefficient? In that case, would seem to be incorrect to call this a kinematic viscosity, in my opinion (even if units are the same). If you are using a diffusion (dispersion) coefficient, which is often based off of a Reynolds number decomposition/gradient diffusion assumption, would also not call this a Reynolds number. Perhaps there is some modifier one can put in front of "Reynolds number", to distinguish it from the usual one. Similarly, wouldn't say this ratio is measuring a transition to turbulence. The flow is turbulent down to a scale of about 1mm (per inertial cascade, to Kolmogorov number). Though I'm not familiar with this "Reynolds number" literature, would assume that this ratio gives some indication of the likelihood of forming large, quasi-2d vortices (what you are calling 'turbulence') vs. having those vortices broken up by dispersive processes (turbulence, shear dispersion, chaotic dispersion....).

This is a fair comment. Our value is actually an effective horizontal diffusivity, and 100 is a reasonable value for that. The text has been updated to state this, but we still have not changed the name of the Re parameter in line with the cited literature. The paragraph now reads "We will use some basic fluid-flow parameters. Transition to turbulence can be parameterised in terms of the Reynolds number, Re, defined as Re = UD/v, where U is a velocity scale, D is the size of the object, and v~100 is a horizontal diffusivity (see, e.g., Wolanski et al., 1984 for details). It indicates when there is a transition to flow separation behind the island: at low Reynolds numbers, Re<1, the flow is quite symmetric upstream and downstream, and there is no flow separation at the object. As the Reynolds number is increased to the range 10 < Re <40, laminar separation happens and results in two steady vortices downstream. As Re increases further, up to Re<1000, these steady vortices are replaced by a periodic von Karman vortex street, whereas if Re>1000, there is a fully separated turbulent flow (Kundu and Cohen, 2002). "

Would note that 2D turbulence is much different than 3D turbulence. The implicit assumption you seem to be making is that once the eddies are formed, they are turbulent. Is this strictly speaking correct? The aspect ratio (horizontal to vertical) of these eddies must be very large, where-as in well-developed turbulence energy should be distributed evenly in x,y, and z (not possible due to continuity in a large eddy in a shallow sea). What is the aspect ratio? Might be good to explore and discuss somewhere, and whether it has any implications for the results. How is the evolution of a 2D eddy different from a 3D eddy? How might bottom friction (or sidewall friction) impact the eddy and make it only quasi 2D? In 3D turbulence, there is a cascade of turbulence from large to small scale. In 2D turbulence, that is not the case—energy transfer goes from small to larger scale (e.g., as when small vortices combine to create a larger one). This is not a paper designed to look at such turbulence issues. However, would be good to be more careful in how turbulence is discussed.

- The aspect ratio is ~10-100, assuming these reach the seafloor (and there is no reason to assume that they shouldn't in a strong barotropic unstratified flow). The eddies act as a sink of tidal energy, which is what we are interested in, and their exact nature is maybe not entirely relevant to this paper. Consequently, we feel that discussing the eddies further will not add to the paper. We note that they are there, and that they complicate the picture of the flow around the island in a way that may not be captured in numerical models, as we already discuss.

Introduction, general: It would be good to briefly review that these small scale 'straits' such as the one being studied are ubiquitous, to frame the larger importance. Angelsey Island in Wales is a nearby example, perhaps. All over the world, there are many Island archipelagos, and some have strong currents such as mentioned here. For example, there is the Greek legend of Charybdis, maybe related to currents through the Strait of Messina (Sicily); see https://en.wikipedia.org/wiki/Strait\_of\_Messina. In Puget Sound, there is Deception Pass (https://en.wikipedia.org/wiki/Deception\_Pass ). Within San Francisco Bay, there is Raccoon Strait. Between New York Harbor and Long Island Sound, there is Hells Gate. There are surely many other examples in the world, and some of them may have been studied or at least have references to large currents and whirlpools. Including some information on or review of them may help frame the broader significance of this study.

Good point, although Anglesey is certainly resolved in the databases we discuss. Also, we do provide a general overview in the introduction that states that any small island will be unresolved, and we are a bit reluctant to bring in further examples, especially since (to our knowledge) this is the first time this type of study – a direct comparison of the effect of a small island compared to altimetry constrained products - has been conducted.

Introduction, general: A brief review of diffusion and dispersion might help frame the "viscosity" you use in your "Reynolds number" (assuming my interpretation above is correct). What is shear dispersion, and is it potentially important here (see for example the book by Fischer et al, from 1979)? What is chaotic dispersion, and is it important here (see Zimmerman, 1986, and de Swart et al., 1997)? How can a jet or plume cause horizontal dispersion (e.g., Fong & Stacey, 2003)? What is turbulent diffusion, and is it important here (usually, it's smaller than shear dispersion caused by lateral velocity gradients, but it also depends on the time scale you are considering—shear dispersion becomes effective at larger time scales than turbulent eddy viscosity (and so on).

- Another fair point, albeit see our response above. The aim of the paper is to investigate how the island change the tides, particularly the tidal stream, and discuss wider implications of those changes. The wake effects were added as an example of an effect that may not necessarily be considered in a modelling or altimetry constrained investigation, but it is not the aim to discuss those in detail, and what has been suggested is most likely not important for the tide but a consequence of the tide. Consequently, we opt to not add any more literature.

Introduction, general comment 3: You could also review the "shallow turbulence" literature, which seems like it might be relevant here. Uijttewaal & Booij, 2000 and Uijttewaal & Jirka, 2003 discuss a "shear stability parameter". Uijttewaal & Booij, 2000 find that eddies produced by lateral shear (du/dy) become increasingly suppressed by bottom boundary layer turbulence as depth decreases. They find that the growth of lateral shear-induced eddies is limited when their shear-stability is greater than approximately 0.1. Again, it should be noted that 2D turbulence is quite different than 3D turbuluence. This generally it isn't much considered in shallow coastal waters, or at least I haven't come across it very much. But maybe there is some more literature since I last thought about it.

See comment above. We agree that 2D and 3D turbulence are very different, but it is again not the point of this paper to discuss that.

Line 89—Did you adjust your pressure measurements for atmospheric pressure variations? If you didn't, would probably be a good idea to do so, just to be complete and make sure that it doesn't significantly alter your analysis. This is particularly true in your "phase 2" result, in which there was a hurricane.

- No, because it doesn't matter for tides; the harmonic analysis is frequency specific and will ignore the storm effects; we have added "The effects of the storm were not noticeable in the tidal signals, as they were at very different natural frequencies. The subsurface pressure

measurements at Bardsey include atmospheric pressure variations, and any tidal variation therein. However, at these latitudes the atmospheric pressure S2 variations are very small. At the equator the atmospheric S2 has an amplitude of about 1.25 mb, which decreases away from the equator as  $[\cos]^{3}$  (latitude), so at 530 N the amplitude is reduced to 0.26 mb, a sea level equivalent of 2.5 mm. In Table 2 the three constituents listed are the two biggest, M2 and S2, and (as an indicator of the presence of shallow water tides) M4, the first harmonic of M2. These shallow water effects are enhanced around the island because of curvature on the directions of current flow."

Line 91 "were subjected to harmonic analysis"—sounds like something unpleasant. Maybe rephrase, e.g., "were harmonically analyzed"?

Disagree, this is a quite standard phrasing in the tidal literature and we have kept our original phrasing.

Line 96: "residuals have standard deviations appropriate for the region"—this is vague. Maybe be specific, and compare it to the nearest tide gauge from the same period.

True, rectified: "The non-tidal residuals, the final column in Table 1, compare well with the residuals at Holyhead, the nearest permanent tide gauge station some 70 km north; for Holyhead these were 0.096 m, 0.172 m, and 0.067 m for the same periods (note that bottom pressure measurements at Bardsey include a partial natural sea level compensation for the inverted barometer effect)."

Line 99—" consistency in the tidal ages" --it might be good to be more specific and define what is meant by 'tidal age', since not all are familiar with this terminology. Is discussion of tidal age needed? Some more specificity on what is considered a good fit would help. Is a good time variation 10 minutes? 1 hour?

**True again, we now define tidal age and we are more specific in the quantification.** "The tidal age is the time after maximum astronomical tidal forcing and the local maximum spring tides, or approximately the phase difference between the phases of  $S_2$  and  $M_2$  in hours,..."

Line 109—Does the TPX09 product use the best altimetry product for near coastal areas? Again, I think JPL has a coastal data product. Would constituents based off of a coastal data product provide better answers? One of the main conclusions in the paper is that satellite data have issues in small scale regions. Is this true of all data products, or just the one used to create the constituent atlas? Another way of putting this—are there other issues, besides resolution, that impact coastal constituents and therefore your comparisons?

Fair question, and there are corrections recently developed that can do a better job at the coast. However, the underpinning numerical model can still not be run globally at enough resolution and the point in the paper is that the global databases – FES and TPXO – are used a lot for tidal work and we want to highlight the issues that may lead to. But, since there are other products that may be better, we have rewritten the introduction: "Scientific understanding of global tidal dynamics is well established. Following the advent of satellite observations, up to 15 tidal constituents have been mapped using altimetry constrained numerical models, and the resulting products verified and constrained further using in situ tidal data – see Stammer et al. (2014) for details. There is, however, still an issue in terms of spatial resolution of the altimetry constrained products: even the most recent (global) tidal models have only 1/300 resolution (equivalent to ~3.2 km in longitude at the equator, some 1.9 km in the domain here, and 3.2 km in latitude everywhere). The satellite themselves may have track separation of 100s of km (Egbert and Erofeeva, 2002) and the coastline can introduce biases in the altimetry data. This means that smaller topographic features and

islands are unresolved, and may be "invisible" in altimetry constrained product even if the features may be resolved in the latest bathymetry databases, e.g., the General Bathymetric Chart of the Oceans (https://www.gebco.net/). This can mean that the energetics in the products, and in other numerical model with insufficient resolution, can be biased because the wakes can act as a large energy sink (McCabe et al., 2006; Stigebrandt, 1980; Warner and MacCready, 2014). Whilst the globally integrated energetics of these models is consistent with astronomical estimates from lunar recession rates (Bills and Ray, 1999; Egbert and Ray, 2001), the local estimates can be wrong. However, new correction algorithms improve the satellite data near coasts (e.g., Piccioni et al., 2018), but this is yet to be included in global tidal products."

General comment: Would be good to establish somewhere what the typical tidal range in this region is, and that diurnal tidal components are small. This will help justify the use of only 4 constituents. (Also, is the use of M4 important? Would be good to establish that quarterdiurnals are important here (or are they)?

Good comment, Section 3.1 now opens woith "A spring-neap cycle of parts of the data from the East and West gauges in Phase 1 is plotted in Figure 2 and show a tidal range surpassing 4 m at spring tide. Note that the diurnal constituents are not discussed further due to their small (<0.1 m) amplitudes. The TG data show M2 amplitudes of 1.210 m (North), 1.347 m (East) and 1.139 m (West, see Table 2)."</li>

Quarterdiurnals are important and included in Table 2.

Line 114—Would define "Highest and Lowest Astronomical tide" (HAT and LAT), before stating that M2+N2 + S2 +M4 are a limited form. Also, strictly speaking, M2 and M4 are phase locked, i.e., 2\*phaseM2 – phase\_M4 = constant (see e.g., Friedrichs & Aubrey 1988). Unless they have a relative phase of zero, it is incorrect to add their amplitudes together to produce HAT. Or, rather, one should consider the relative phase when adding. Is that done here?

This is done, but in a different way. We now say "The altimetry constrained product used in this paper is that of the TPXO9 ATLAS which is derived from assimilation of both satellite altimeter and tide gauge data (Egbert and Erofeeva, 2002). The resolution is 1/30° in both latitude and longitude (3.7 km and 2.2 km at Bardsey). We used the elevation and transport information, and their respective phases, for the M2, S2, and M4 constituents. In the following calculations, we approximate the largest tidal current speeds or amplitudes as the sum of the amplitudes of the above three tidal constituents. Of course this is only a crude estimate of the full Highest and Lowest astronomical tides. Note that we are not allowing for M2 to M4 phase locking, and the relatively small diurnal tides are ignored. We refer to this as the GA (Greatest Astronomical) in the following."

Line 117—This is the first mention that I can recall of Landsat. Why are these images being downloaded? Leading with a topic sentence that provides some context would be good.

Done: "Landsat-8 data images were used to identify possible eddies in the currents and further illustrate unresolved effects due to the island. Note that we are not aiming for a full wake description in this paper. Data were downloaded from the Earth Explorer website (https://earthexplorer.usgs.gov/). True colour enhanced RGB images were created with SNAP 7.0 (Sentinel Application Platform; https://step.esa.int/main/toolboxes/snap/) using the panchromatic band for red (500 - 680nm, 15m resolution), band 3 for green (530 - 590nm, 30m resolution) and Band 2 for blue (450 - 510 nm, 30m resolution). The blue and green bands were interpolated using a bicubic projection to the 15m panchromatic resolution, and brightness was enhanced to allow easier visualization of the wakes. The images used were taken between 11:00 and 12:00 UTC, when the satellite passed over the area, and the two

images were the only cloud-free ones during the measurement periods that were on different stages of the tide."

Line 129—The results lead with a table. I would have expected some text before a table. Maybe put the table elsewhere?

- Done.

Amplitudes and phases—Can you think of some way to report confidence intervals or uncertainty, beyond the statement about significant figures?

The text has been updated to include "The non-tidal residuals, the final column in Table 1, compare well with the residuals at Holyhead, the nearest permanent tide gauge station some 70 km north; for Holyhead these were 0.096 m, 0.172 m, and 0.067 m for the same periods (note that bottom pressure measurements at Bardsey include a partial natural sea level compensation for the inverted barometer effect)."

Line 145—what about frictionally produced overtides? With a strong current, would seem likely.

- We analyse for them – see M4 in the table and the text above. There are more, but they are small.

Line 148-151—The use of numbers could be reduced and the point made more succinctly, here and elsewhere. For example, you could say that TPX09 data suggests only a 0.02m and <1 degree difference in M2 in the cross-channel direction, compared to ~0.19m and 6.5 degrees with in-situ data (see Table xxx). A reader can look at the table for the exact numbers, but doesn't necessarily need to know the exact numbers in the narrative arc (or rather, only needs to know that the TPX gives a much different, and less correct, answer).

Thank you for the suggestion. This paragraph now reads "We turn now to a comparison of the tidal analysis data for M2 from the two sources (see Table 2 for details). When the TPXO9 M2 data, which has no Bardsey island representation, is interpolated linearly to the TG positions, the result is only a 0.02 m and 0.7° amplitude and phase difference for the Phase 1 locations. Compared to the 0.19 m amplitude difference and 6.5° phase difference in the TG data, it is obvious that there is a substantial deficiency in the TPXO9 model in representing the role of the island due to its limited resolution. These results are supported by the Phase 2 measurements (Table 2). Phase 3 saw an extended and different approach to the data collection. We revisited East, but also deployed two gauges on the Llŷn peninsula, on the approach to the island (South Mainland)), and north of it (North Mainland). At South Mainland, TPXO is again underestimating the tidal amplitude by more than 10%. At North Mainland, some 5 km north of Bardsey, and just north of the Sound, however, the TG and TPXO amplitudes are within 1 cm of each other. This again shows the effect Bardsey and local topography have on the tidal amplitudes in the region."

Line 162-168: For someone not familiar with this area, the heavy use of place names is sometimes confusing.

- We provide a map with the place names and are not quite sure how we would describe what is happening without using them. We could perhaps label areas "a", "b" etc, but the locations have names found on a map and we will stick with those.

General comment:

Can one be sure that estimates of M2 and M4 from TPX09 are directly comparable to your one month long measurements, given things like seasonal and interannual variation? Some discussion and exploration would be good. It seems to me that some review of the TPX analysis would help one frame

the results, and help rule out environmentally-based factors as the source of differences in the constitutent analysis. What is the sampling rate of TPX data, and how long of a data set is needed to obtain good estimates of M2, M4 etc? Since a long time period is needed, any seasonal variation in tidal constituents are averaged out (see e.g. one of the Mueller papers, or Graewe et al. 2014, or others) . However, the in-situ data would be effected by seasonal effects, and possibly astronomical factors such as the strength of the spring-neap cycle over the measurement month (through frictional interaction). Meteorlogical events like the afforementioned hurricane could also affect M2 and M4, possibly. One way to look at seasonal cycles would be to evaluate the seasonal cycle in M2 at the nearest long-term tide gauges. Does such an analysis suggest this a factor in the comparison with TPX? A seasonal cycle in M2 would produce an M4 variability as well, and therefore any comparison with TPX. In shallow water, my experience is that M4 can vary a lot from year to year. TPX constituents are measured over many years, and may therefore "average over" interannual variability. Other environmental/astronomical variability could also be excluded as a potential factor in your comparison. Does TPX consider the nodal cycle? Do you adjust for the nodal cycle in in-situ data?

The nodal cycle in the M2 tide is averaged over in TPXO, and as we state on line 178 in the caption to table 2 our observational values have been adjusted for nodal and seasonal effects. M4 changes with the nodal cycle too, of course, and is again averaged over in TPXO and adjusted for here. The text has been updated to make this clearer: "Amplitudes and phases of tidal constituents based on short periods of observations need adjusting to reflect the long term values of amplitudes and phases. The values in Table 2 have been adjusted for both nodal effects and for an observed non-astronomical seasonal modulation of M2. Standard harmonic analyses include an automatic adjustment to amplitudes and phases of lunar components to allow for the full 3.7%, 18.6 year modulation due to the regression of lunar nodes. However, the full 3.7% nodal modulation is generally significantly reduced in shallow water and shelf seas, so locaL counter adjustments are needed. The nodal M2 amplitude modulation at Holyhead, the nearest standard port, is reduced to 1.8% (Woodworth et al., 1991). We have used this value in correcting the standard 3.7% adjustment. The M4 nodal modulations are twice that for M2. The seasonal M2 modulations are generally observed to have regional coherence, so we have used the seasonal modulations from 9 years of Newlyn data (in the period 2000-2011). M4 is not seasonally adjusted, and S2 is not a lunar term, so is not modulated nodally. These very precise adjustments are possible and useful, but overall as stated in the caption to Table 2, for regional comparisons we assume, slightly conservatively, confidence ranges of 1% for amplitudes and 1.0 degrees for phases."

Line 189, Equation 189—What about frictional effects? Would seem that a fudge factor might be warranted, or perhaps a scaling symbol rather than an equal sign. In any case, friction is important, and would be good to account for somehow.

- True, and the revised text discusses this. We don't account for it, but rather have made the whole discussion shorter (see reply to RC1).

Line 191-202—Seems like this paragraph could be reduced in size/explained more succinctly - Indeed, edited (see reply to RC1 above).

Line 191-202 - M2 is being used in the scaling equation (Equation 1) and is being compared to a vague maximum velocity of 4m/s. However, wouldn't the maximum velocity be more likely during a high spring tide, i.e., when the tidal amplitude is caused by M2 +S2 +N2? Ok, I see this is in the next paragraph. However, am leaving this comment in, because this paragraph and the next could be presented more succinctly, perhaps together. Also, would suggest seeing if there are any model or insitu results in the peer-reviewed literature than provide estimates of the velocities in this strait, and/or the actual measurements which form the basis for the admiralty charts. The '4 m/s' maximum velocity

is quite vague, and the context of this measurement is unknown (was it a wind day? Is it a point measurement, or depth/width averaged? Etc, etc). Therefore, using this value as the gold standard for comparison is a bit iffy.

Thank you, this has been rewritten (see reply to RC1 as well). As for the current speed: it comes from the admiralty chart and discussions with the local fisherman who lives on the island. There are no other current measurements from the strait and it will have to do alongside our new current estimate.

Line 224—Ok, I see now that friction is being considered. Maybe it would make sense to include all the theory in the Methods section, so that it is more clear that you are considering frictional effects? Note there is no Equation 2 in the manuscript (i.e., Equation 3 is ms-labeled).

- The numbering has been corrected. The friction discussion is covered in the replies mentioned above.

Line 226—Did you check that the phase speed really is sqrt(gh)? Since you have the phase progression and know the depth, would be good to check. In shallow water when there is friction and/or convergence, the phase speed can be quite different than sqrt(gh). See e.g., Jay 1991.

This is true, but it is the group speed that transports energy, not the phase, and the group speed for shallow water waves remain sqrt(gh) as far as we are aware.

Line 226-234—How does this dissipation estimate compare to more local estimates of dissipation, e.g., within the region between England/Wales and Ireland?

- The total dissipation on the European Shelf is about 180 GW (Egbert and Ray (2001), so it is a small part of that. In Liverpool Bay, you find similar current speeds so the dissipation rates will be similar. In the Irish sea, the currents are about 0.5 m/s, so the dissipation rates will be far smaller. The latter part of the paragraph now reads "The dissipation in a tidal stream can also be computed from  $\varepsilon = \rho C_D |u|^3$ , where Cd~0.0025 is a drag coefficient (Taylor, 1920). Using the TPXO9 current speed in the strait, assuming the Sound to be 3.1 km wide and 2 km long, the GA spring dissipation comes out as 53 MW (using u=1.5 m s-1), and the M2 dissipation (using a current speed of 1.2 m s-1) as 28 MW. This is a substantial underestimate compared to the estimates above (factors of 7 and ~4.5 for the GA and M2 tides, respectively), which again highlights the importance of resolving small-scale topography in local tidal energy estimates, and the use of direct observations in coastal areas to constrain any modelling effort. This dissipation here is only a small fraction of the European Shelf and coastline, but it is a very energetic area. Although the Bardsey tides are unusually energetic, underestimated local coastal energy dissipation may be substantial in the TPXO9 (and similar) data and numerical models.

Image analysis—how many images were looked at? How representative and statistically significant is the analysis? I would consider looking at more images, to see if the qualitative results are repeatable. For example, you could look at Landsat 7 or Landsat 5 data. You might also consider looking at the ESA Sentinal-3 data as well. It has fantastic resolution and better time resolution than Landsat.

- There are unfortunately no more images during the measurement periods that are at different stages of the tide and with clear skies. Yes, there are better products, but this is not the main focus of the paper and they are added to show that there are effects even behind small islands that many models will not catch, with potential wider implications. We have highlighted this in the paper: "Landsat-8 data images were used to identify possible eddies in the currents and further illustrate unresolved effects due to the island. Note that we are not aiming for a full wake description in this paper. Data were downloaded from the Earth Explorer website (https://earthexplorer.usgs.gov/). True colour enhanced RGB images were

created with SNAP 7.0 (Sentinel Application Platform; https://step.esa.int/main/toolboxes/snap/) using the panchromatic band for red (500 - 680nm, 15m resolution), band 3 for green (530 - 590nm, 30m resolution) and Band 2 for blue (450 - 510 nm, 30m resolution). The blue and green bands were interpolated using a bicubic projection to the 15m panchromatic resolution, and brightness was enhanced to allow easier visualization of the wakes. The images used were taken between 11:00 and 12:00 UTC, when the satellite passed over the area, and the two images were the only cloud-free ones during the measurement periods that were on different stages of the tide."

General comment: You might consider looking at Pawlak & MacCready 2001 and Warner & MacCready 2014 for discussion of form drag and eddy formation in the wake of small-scale topography in Puget Sound. Though a stratified region, there might be some useful insights or results in those papers. They also use the Bernoulli Equation, but consider the time-varying potential as well.

- Thank you, we have added more references to the general discussion in the introduction: "This can mean that the energetics in the products, and in other numerical model with insufficient resolution, can be biased because the wakes can act as a large energy sink (McCabe et al., 2006; Stigebrandt, 1980; Warner and MacCready, 2014).".
  - We do not expand on these further since the aim is to see how wrong the altimetry constrained products are.

General comment: Some more explanation of global models and their resolution is needed. Why is dissipation an important issue? Making this connection will help prove the point that smaller scale resolution can be important.

- **Added:** "Whilst the globally integrated energetics of these models is consistent with astronomical estimates from lunar recession rates (Bills and Ray, 1999; Egbert and Ray, 2001), the local estimates can be wrong."

---

## Author Comment (AC4) · 19 Aug 2020

A clarification: the replies are in the supplementary material, not inteh text box.

---

## Author Response (AR2)

Reply to Dr William's editor comments
Thank-you for the much improved manuscript and response to reviewers. I don't think it needs to go
back to the reviewers as you have largely addressed their comments, however there are a few
remaining minor issues.
**- Thank you for your constructive comments and efforts handling this paper. We have addressed**
**the remaining concerns below and hope it us now suitable for publication**
Fig 1b - TG locations (plural).
Regarding the use of place names, Aberdaron is marked on the map but not necessary to the text, and
the Llyn Peninsula isn't on the map (Fig 1). I suggest you remove one and add the other to the map.
-   **Done**
line 106 These shallow water effects are enhanced around the island because of curvature on the
directions of current flow. I don't understand this. Rephrase?
-   **Rewritten: "**Shallow water tides  are enhanced around the island because of the curvature
of the flow as it bypasses the island and headland (see section 6.2.3 of Pugh and
Woodworth, 2014).**"**
Caption of Table 1 refers to tidal phase but there's not phase info in that table.
Table 1 - Better not use dd/mm/yy for dates at all, it's confusing for Americans (though at least it's in
the caption). Apparently the house standard is 25 July 2007 (dd month yyyy), 15:17:02 (hh:mm:ss) .
yyyy-mm-dd HH:MM is probably also OK if you're short of space.
-   **Removed the phase (and amplitude) reference and changed date format as suggested.**
Throughout: I suggest getting rid of "Phase" to refer to the measurement campaigns and replace by
"Deployment" or similar. Otherwise there's phrases like "phase 1 residuals" which is confusing for no
good reason.
-   **Good point, replaced with "**Deployment**" throughout.**
The data is available to download as required. Thank-you. You might consider also depositing it in a
suitable respository of similar data.
-   **This is under consideration.**
line 184: I really dislike "obviously", it's either patronising or a wallpaper-word. I guess it's there to
placate those reviewers to whom it's obvious.
-   **We have obviously removed this; we agree that it can be patronising.**
line 204-208: Fig 3 has red & black curves? & level differences are not plotted, so line 208 is not right.
-   **Corrected; it now reads "**Figure 3 a and b (red curves) shows the currents so computed, for
Day 147 (spring tides) and Day 154 (neap tides), with the speeds are in metres per second.
The black curves are the measured sea levels at East. … The noise in the level differences,
which appears as noise in the currents (i.e., the red curves), may be an indication of
turbulence and eddies discussed further below.**"**

Fig 4 a and b : It's not ideal that the coloured cells are aligned with the arrows on their corners rather
than centred, though I know it's a bit awkward to correct in matlab. It means the green cells are
misalinged with the longest arrows. And it makes me wonder if it flags a possible problem as matlab
will default to plotting interpolated data rather than the actual values of each cell. Please check that
the plot is as intended. It is important as it affects the argument about TPXO9 success in replicating
the tide. Also please edit the colour scale so it doesn't saturate at 2.5, so we can see how high it gets.
(c) has a of maximum only about 1.6 - which cell is represented? Perhaps draw a box round those
cell(s)? Why is it less than 2.5 from the colour scale?
- **The arrow positions are corrected to the centre of the cells – this is indeed an issue and we**
**appreciate having it pointed out.**
**The shading here isn't interpolated but "flat", so the actual data is shown (hence the**
**"blocky" structure). We have not changed that.**
**The colourbar no ends at 3.5 m/s**
**The data in (old) panel c/now panel d is from the actual sound and not from the maximum**
**cells – that is indeed confusing and the cell has been marked in the new panel a. Th reason**
**for this is that our computations based on the TG data are valid in the Sound itself, hence a**
**comparison to that box. The new figure and caption are now:**

[Figure]

Figure 1:  a) The depth from the TPXO9-database covering Bardsey (marked with a white open circle).
The rectangle north-west of the island shows the grid cell the data in panel d was extracted from.
a)-b) The current magnitude (colour) and vectors at neap (a) and spring (b) flood tides from TPXO9.
These are computed from the M2 and S2 constituents only. The white circle shows the location of
Bardsey – note that it is not resolved in the TPXO9 data and has been added for visual purposes only.
d) The magnitude of the tidal current during a spring-neap cycle in the Sound (i.e., at the cell marked
with a rectangle in panel a) using the M2, S2, and M4 constituents in the TPXO9 data. Note that we
chose to show data from the centre of the Sound because that is where the computations using Eq.
(1) are valid.

- **We have added a panel of this to figure 4.**

lines 284-306: You've used one method to calculate energy from obs, and another for the energy from

TPXO9? It makes it hard to compare... can you use the same for both? Also there's some repetition here.

**OK, fair point and a remnant from before the estimate of the currents using Eq. 1. We now only use**

**the direct dissipation computation, which proves the point even further. The section now reads "**The dissipation in a tidal stream can also be computed from $\varepsilon = \rho C_D |u|^3$, where Cd~0.0025 is a drag coefficient (Taylor, 1920) and $\rho$=1020 kg m$^{-3}$ is a reference density. The peak dissipation using the computed GA current data from Eq. (1) and shown in Figure 3 gives 777 MW for springs and 426 MW

for neaps, assuming the sound is 3.1 km wide and 2.2 km long. This is 0.2-0.4% of the 180 GW of $M_2$

dissipation on the European shelf (see Egbert and Ray, 2000), and is a reasonable estimate for such an energetic region. Note that this method is independent of the phases between the locations, nor does it depend on the phases between the amplitudes and currents. If we instead use the  the TPXO9

current speed in the strait, the GA spring dissipation comes out as 53 MW (using u=1.5 m s$^{-1}$), and the

$M_2$ dissipation (using a current speed of 1.2 m s$^{-1}$) as 28 MW. This is an underestimate of a factor 14

for the GA spring tide compared to the computation from the TG data, which again highlights the importance of resolving small-scale topography in local tidal energy estimates, and the use of direct observations in coastal areas to constrain any modelling effort. This dissipation here is only a small fraction of the European Shelf and coastline, but it is a very energetic area. Although the Bardsey tides are unusually energetic, underestimated local coastal energy dissipation may be substantial in the

TPXO9      (and      similar)      data      and      numerical      models.**"**

line 319 : astronomic speed . Speed during GA tide?

- **Corrected**

line 350: marked with arrows? These have now gone? If you do add more, can I suggest magenta, for better contrast with clouds.

- **They have indeed gone AWOL, added back in:**

[Figure]

-

Table 2 would be less cluttered and easier to read if you got rid of most of the lines. "Horizontal lines should normally only appear above and below the table, and as a separator between the head and
the main body of the table." Also some of the vertical lines - I suggest you group in pairs.
- **We have tidied the table and left what we think are necessary lines to aid the reading. We**
**are aware that we have more than the norm, but it is a complex table and removing more**
**would increase the risk of confusing the reader.**

| Station | | M2 | | S2 | | M4 | | Tidal Age (hours) | M2/S2 ratio |
|---|---|---|---|---|---|---|---|---|---|
| | | TG | TPXO | TG | TPXO | TG | TPXO | | |
| DEPLOYMENT 1 | | | | | | | | | |
| North | H | 1.210 | 1.17 | 0.458 | 0.45 | 0.114 | 0.12 | | 0.378 |
| | G | 250.4 | 254.4 | 287.1 | 287.3 | 21.7 | 32.4 | 36.66 | |
| East | H | 1.326 | 1.16 | 0.514 | 0.42 | 0.147 | 0.12 | | 0.387 |
| | G | 245.6 | 253.8 | 283.4 | 286.7 | 49.7 | 34.3 | 37.76 | |
| West | H | 1.139 | 1.15 | 0.434 | 0.42 | 0.138 | 0.12 | | 0.381 |
| | G | 252.1 | 253.7 | 288.4 | 286.6 | 36.1 | 34.8 | 36.26 | |
| DEPLOYMENT 2 | | | | | | | | | |
| NW | H | 1.159 | 1.16 | 0.431 | 0.42 | 0.132 | 0.12 | | 0.372 |
| | G | 254.2 | 254.7 | 287.1 | 287.6 | 36.4 | 33.4 | 32.88 | |
| SW | H | 1.217 | 1.15 | 0.461 | 0.42 | 0.09 | 0.12 | | 0.379 |
| | G | 251.2 | 253.4 | 285.5 | 286.3 | 27.4 | 35.6 | 34.28 | |
| NE | H | 1.271 | 1.15 | 0.482 | 0.43 | 0.096 | 0.12 | | 0.379 |
| | G | 250.4 | 253.8 | 284.0 | 286.7 | 44.0 | 32.8 | 33.58 | |
| DEPLOYMENT 3 | | | | | | | | | |
| East | H | 1.351 | 1.16 | 0.522 | 0.42 | 0.138 | 0.12 | | 0.386 |
| | G | 247.3 | 253.8 | 282.8 | 286.7 | 55.0 | 34.3 | 35.5 | |
| S. Mainland | H | 1.397 | 1.21 | 0.538 | 0.44 | 0.152 | 0.14 | | 0.385 |
| | G | 245.1 | 251.5 | 280.7 | 284.4 | 51.7 | 37.1 | 35.6 | |
| N. Mainland | H | 1.228 | 1.2 | 0.461 | 0.43 | 0.074 | 0.12 | | 0.375 |
| | G | 257.2 | 254.6 | 290.4 | 287.6 | 40.8 | 29.1 | 33.2 | |

-
You have bypassed reviewer 2's question on uncertainty on tidal constituents. I know why, it's not
calculated in TASK, and is not trivial, but I think you do need to comment a bit further on this.
The natural place for the non-tidal standard deviation would be in Table 2 with the results rather than
Table 1. What is the non-tidal residual standard deviation of the observations if TPXO9 tides are
assumed correct? What if only M2+S2+M4 (from each) is used (ie exactly how much is omitted in all
the ignored constituents?)
- **We do include an estimate of this in the caption for table 2, and the non tidal variance is in**
**table 1. To clarify, the following text has been added to the opening of section 3: "The**
results of the tidal harmonic analyses are shown in Table 2. The *in situ* RBR data results are
given to 0.001 m and 1.0 degrees. Amplitudes are given to three decimal places as appropriate
for the uncertainties in the RBR data, whereas the timing of constituent phases is probably better
than 0.5º (1 minute in time for $M_2$). Given the small local tidal differences, it is necessary to
consider possible variability among the RBR tidal constituents across the three deployments,
both due to seasonal, and also due to nodal shifts. Also, there is a statistical uncertainty against
background noise, as discussed in Pugh and Woodworth, 2014, Section 4.6. This statistical
uncertainty depends on the estimate of non-tidal noise across the semidiurnal tidal band, though this can be optimistic as noise may be more sharply focussed at the M2 frequency. In fact, the
seasonal uncertainty is most significant here. Based on uncertainties in making the seasonal and
nodal adjustments we conclude that, for regional comparisons we can assume confidence ranges
of 1% for amplitudes and 1.0 degrees for phases. We also note that for station East in 2017,
M2+S2+M4 (i.e., our GA) accounts for 93.6% of the tidal variance, with N2, in fourth place,
provides 3.7% of the remainder.**"**

And finally, there's a lot of typesetting problems. I think the typesetter will help with these at the next
step but please help by setting the equations and variables in italics where appropriate, and ensuring
tidal constituents are correctly set. m s-1. Etc. Please check the proofs carefully.
- **Thank you, we have proofed the text again and will read the final proofs carefully.**

[revised manuscript text omitted]